# Carbonyl sulfide measurements from a South Pole ice core and implications for atmospheric variability since the last glacial period

Murat Aydin[1], Melinda R. Nicewonger[1], Gregory L. Britten[2,3], Dominic Winski[4], Mary Whelan[5], John D. Patterson[1], Erich Osterberg[6], Christopher F. Lee[1,7], Tara Harder[1], Kyle J. Callahan[1,8], David Ferris[6], Eric S. Saltzman[1]

1Department of Earth System Science, University of California, Irvine; Irvine, CA USA.

2Biology Department, Woods Hole Oceanographic Institute of Technology, Cambridge, MA USA.

3Department of Earth, Atmospheric, and Planetary Sciences, Massachusetts Institute of Technology, MA USA.

4Climate Change Institute, University of Maine; Orono, ME USA.

5Rutgers Climate Institute, Rutgers University; New Brunswick, NJ USA.

6Department of Earth Sciences, Dartmouth College; Hanover, NH USA.

7Department of Chemistry and CIRES, University of Colorado Boulder, CO USA.

8Department of Physics and Astronomy, University of California, Los Angeles, CA USA.

*Correspondence to*: Murat Aydin (maydin@uci.edu)

**Abstract.** Carbonyl sulfide (COS) is the most abundant sulfur gas in the atmosphere with links to terrestrial and oceanic productivity. We measured COS in ice core air from an intermediate depth ice core from the South Pole using both dry and wet extraction methods, recovering a 52,500-year record. We find evidence for COS production in the firn, altering the atmospheric signal preserved in the ice core. Mean sea salt aerosol concentrations from the same depth are a good proxy for the COS production, which disproportionately impacts the measurements from glacial period ice with high sea salt aerosol concentrations. The COS measurements are corrected using sea salt sodium (ssNa) as a proxy for the excess COS resulting from the production. The ssNa-corrected COS record displays substantially less COS in the glacial period atmosphere than the Holocene and a 2-4 fold COS rise during the deglaciation synchronous with the associated climate signal. The deglacial COS rise was primarily source driven. Oceanic emissions in the form of COS, carbon disulfide (CS$_2$), and dimethylsulfide (DMS) are collectively the largest natural source of atmospheric COS. A large increase in ocean COS emissions during the deglaciation suggests enhancements in emissions of ocean sulfur gases via processes that involve ocean productivity, although we cannot quantify individual contributions from each gas.

## 1 Introduction

The sources of atmospheric carbonyl sulfide (COS) include anthropogenic activities, oceans, biomass burning, anoxic ecosystems, and volcanism (Berry et al., 2013; Kettle et al., 2002; Whelan et al., 2018). Present-day global mean levels are 480-490 pmol mol$^{-1}$ (ppt) (Montzka et al., 2007), with anthropogenic emissions accounting for about 40% of emissions (Table 1). The oceans are the largest natural source, which include direct emissions in the form of COS and indirect emissions in the form of precursor gases carbon disulfide (CS$_2$) and dimethylsulfide (C$_2$H$_6$S or DMS) whose atmospheric oxidation products

include COS (Berry et al., 2013; Kettle et al., 2002; Jernigan et al., 2022; Lennartz et al., 2017; 2019; Whelan et al., 2018). In the ocean, COS and $CS_2$ are produced primarily by photochemical reactions involving organosulfur compounds (Kettle et al.,

2002; Lennartz et al., 2017; 2019; Whelan et al., 2018), and DMS is a byproduct of phytoplanktonic activity (Andreae, 1990; Charlson et al., 1987; Ksionzek, et al., 2016; Lana et al., 2011; Vogt and Liss, 2009; Wang et al., 2018a). Warmer waters can act as a seasonal sink due to temperature dependent loss to hydrolysis, but the world oceans are a large net source on an annual average basis with respect to both the direct and indirect emissions (Kettle et al., 2002; Lennartz et al., 2017; 2020). Biomass burning, anoxic wetlands, and volcanism are considered minor sources (Berry et al., 2013; Kettle et al., 2002; Montzka et al.,

2007; Whelan et la., 2018). The primary removal mechanism of atmospheric COS is uptake by terrestrial plants during photosynthesis, which accounts for 70-80% of total atmospheric removal of COS (Table 1). The remaining losses are attributed to soil uptake, atmospheric oxidation via OH, direct photolysis in the atmosphere, and stratospheric loss, resulting in a tropospheric lifetime of 2-3 years (Berry et al., 2013; Kettle et al., 2002; Montzka et al., 2007; Whelan et al., 2018).

Past atmospheric variability of COS is of interest because COS is a precursor for background stratospheric sulfate

aerosol with impacts on stratospheric chemistry and a net negative radiative impact (Kremser et al., 2016; Quaglia et al., 2022; Sheng et al., 2015; Solomon et al., 1999). Further, atmospheric COS is associated with oceanic production and emission of DMS, which is a major source of marine sulfate aerosol, inducing negative direct and indirect climate feedbacks (Boucher et al., 2013; Thomas et al., 2010; Wang et al., 2018b). Polar ice cores contain ancient air that can be used to reconstruct past atmospheric composition of trace gases. Previous measurements of COS in Antarctic ice cores revealed slow, temperature-

dependent degradation of COS in the ice core air due to hydrolysis (Aydin et al., 2014). The kinetic parameters of the in-situ COS loss have been estimated, allowing reconstruction of a 54,000-year composite ice core COS record based on data from multiple Antarctic sites including glacial period measurements from the West Antarctic Ice Sheet (WAIS) Divide and Taylor Dome sites (Aydin et al., 2016).

The existing composite ice core record had multiple limitations (Aydin et al., 2016). First, the measurements from ice

with warmer temperature histories require large corrections for loss. Second, the measurements from the bubble-clathrate transition zone (BCTZ) at the WAIS Divide site displayed large negative biases; the BCTZ refers to the depth range over which the air inclusions in the ice transition from gas phase (bubbles) to air hydrate clathrates (clathrates) due to increasing hydrostatic pressure (Ikeda et al., 1999). Third, the temporal resolution of measurements from either ice core was low over broad age horizons and the chronology of the Taylor Dome ice core was highly uncertain during the glacial period and the

glacial/interglacial transition. While the data from the two ice core sites displayed lower levels early in the glacial period than the Holocene in agreement with each other, they trended in different directions during the deglaciation, complicating the atmospheric interpretation of the record (section 3). Measurements from other ice cores were necessary to determine whether the discrepancies were solely due to unquantified uncertainties associated with the known limitations or if there were unidentified problems such as production processes altering ice core COS levels.

In this work, we present new measurements from the intermediate depth SPC14 ice core recently drilled to 1751 m at the South Pole as part of the SPICEcore project (Souney et al., 2020). These measurements were conducted at considerably

higher resolution than the prior data sets, resulting in a continuous ice core COS record spanning the 52.5 ky (kiloyears) before present (1950 CE), and the cold mean annual surface temperatures at the South Pole (-50°C) mean that measurements do not require a correction for loss to hydrolysis (Aydin et al., 2014; 2016). These characteristics allow evaluation of the SPC14
measurements with respect to possible impurity-related production in the ice sheet, resulting in excess (not of atmospheric origin) COS. Presence of excess COS can impede the recovery of an atmospheric record from the ice core measurements. This is particularly relevant for measurements from glacial period ice, which typically contain high concentrations of ice impurities. Here, we rely on measurements of soluble ions $Ca^{2+}$ and $Na^+$ (Winski et al., 2021) as indicators for land- and marine-sourced ice impurities to check for the presence of excess COS in the SPC14 ice core. We find compelling evidence for presence of
excess COS through the entire SPC14 ice core. The amount of excess COS is quantified and corrected for using linear regression analyses between sea salt Na (ssNa) and COS, resulting in an inferred atmospheric record of COS. We implement a Bayesian approach within a state-of-the-art numerical framework to quantify and propagate the full range of uncertainty arising from the corrections to the atmospheric COS record. The atmospheric record is interpreted rigorously within the uncertainties, relying on the contemporary knowledge of the atmospheric COS budget.

**2 Methods**

**2.1 Dry and wet extraction-based COS analysis**

All COS measurements were conducted at the UCI ice core trace gas laboratory. We analyzed 10-20 cm long samples from the designated gas cross-section of the SPC14 ice core. Air entrapped in the ice core samples is extracted using dry (frozen) and wet (melt) extraction methods that have been previously used for ice core measurements of COS and other trace
gases (Aydin et al., 2007, 2014, 2016; Nicewonger et al., 2016). In dry extraction, ice core air is liberated via mechanical shredding of frozen ice samples in four identical stainless-steel vacuum chambers (6 L), each fitted with a flat, sharp cutting surface. Samples are placed on the cutters inside the vacuum chambers and shredded by linear oscillations (15 cm throw at 1 Hz for 20 min) inside a -50°C freezer. In wet extraction, ice core air is liberated by melting the samples in two identical glass vessels at 30°C which typically takes 30-40 min. The SPC14 dry extraction samples were 400-500 g and wet extraction samples
were 300-350 g. Melting releases more than 99% of the ice core air (Nicewonger et al., 2020). The dry extraction efficiency for the UCI dry extraction vessels is about 60% for bubbly ice samples and 40% for full clathrate ice, with an expected drop in the extracted air amount across the BCTZ (Aydin et al., 2016). The clathrates are smaller in size than bubbles and harder than ice (Ohno et al., 2004; Salamatin et al., 1998); the relatively lower extraction efficiency of the dry extraction method for clathrates versus bubbles is presumed to be primarily due to a combination of these two factors.

In the BCTZ, the gas composition of the air in bubble and clathrate phases can display extreme fractionation likely due to different diffusive fluxes for different gases during the transition process, and at the completion of the transition, the composition of the air hydrates is expected to reflect atmospheric levels (Ikeda et al., 1999; Ikeda-Fukazawa et al., 2001; Ohno et al., 2004). Dry extraction-based measurements from the WAIS Divide ice core suggest dry COS measurements are prone to

low bias and higher scatter in the BCTZ, suggesting the bubbles, which are extracted more efficiently than the clathrates, are depleted with respect to COS (Aydin et al., 2016). The SPC14 data set does not include any dry measurements from the initially anticipated BCTZ zone (900-1200 m); this sampling strategy was designed to avoid the measurement artifacts associated with the low extraction efficiency of dry extraction (Fig. 1a). In the SPC14 dry extractions, the amount of air extracted from the samples start to drop at 800 m (Fig. A1a), suggesting the BCTZ starts about 100 m above the initially anticipated depth of 900 m.

The wet extraction system and methods have been developed for non-methane hydrocarbon measurements (Nicewonger et al., 2016 and 2020) and the SPC14 measurements presented here are the first COS data from the wet extraction system. BCTZ does not represent a problem for the wet measurements because of the >99% efficiency of this extraction method. However, COS is more soluble in water than air, resulting in a negative bias in wet measurements (Fig. 1a) which has to be corrected for (section 2.2).

Following either wet or dry extraction, the sample air is frozen in a stainless-steel tube dipped in liquid helium (4K) for transfer to the analysis inlet system used for pre-concentration of the trace gases followed by GC-MS (gas chromatography-mass spectrometry) analysis. COS is measured with the same analytical instrumentation regardless of the ice core air extraction method. Trace gases including COS are pre-concentrated on a glass-bead trap using liquid nitrogen (77K), followed by transfer onto a capillary trap (77K) before being thermally injected onto a DB-1 GC column. The GC is connected to a dual focusing magnetic sector MS that operates at a mass resolution (m/$\Delta$m) of 8,000-10,000 at 5% peak height. The GC-MS system is calibrated with an isotope dilution technique using a [13]C-labeled COS isotope standard ([13]COS) along with ppt-level unlabeled [12]COS standards. Both labeled and unlabeled ppt-level standards are prepared in our laboratory from ppb-level primary (long term) standards. The primary standards are also prepared in our laboratory from commercially available pure gases. A known amount of ppt-level [13]COS internal standard is mixed with every sample during the pre-concentration step and co-analyzed with the COS from the sample ice core air.

All measurements are reported as dry molar mixing ratios and are corrected for 2% gravitational enrichment based on hydrostatic equilibrium at the present-day firn close-off depth of 120 m at the South Pole. For simplicity, we ignore temporal variability in the gravitational correction. The errors arising from this simplification should be less than 1% of the reported mixing ratios.

## 2.2 Solubility correction for the wet extraction measurements

We calculate the solubility correction empirically for each sample and correct the wet measurements. The ratio of total moles of COS to what is left in the ice core melt water during wet extraction can be calculated by (Nicewonger et al., 2020):

$$\frac{m_t}{m_l} = 1 + \frac{V_g}{hV_l} \tag{1}$$

In Eq. (1), $m_t$ is the total moles of COS, $m_l$ is moles of COS left dissolved in the melt water, $V_g$ is the volume of the gas (headspace) in the melt chamber, $V_l$ is the volume of the melt water, and $h$ is the dimensionless effective solubility (aqueous

concentration divided by gas concentration). We assume $h$ to be the same for every sample melt. $V_g$ can be substituted by volume of the melt chamber ($V_c$) minus $V_l$, and Eq. (1) can be rearranged into expressing $m_t/m_l$ as a linear function of a single variable $V_l$ and two constants ($C_1$ and $C_2$), given that $V_c$ is also constant:

$$\frac{m_t}{m_l} = C_1 + C_2 \frac{1}{V_l} \qquad (2)$$

On the left side of Eq. (2), $m_t$ scales with the dry extraction COS mixing ratio ($COS_{dry}$) and $m_l$ scales with the difference between the dry and wet ($COS_{wet}$) extraction mixing ratios at the same depth, assuming solubility of air in the melt water is negligible compared to that of COS. Nicewonger et al. (2020) estimated 0.7% of the air could be left in the melt water for an average size sample at saturation equilibrium. The dry extraction mixing ratio at a given depth is estimated by linear interpolation from the smoothed dry extraction record (Fig. A2). The $V_l$ on the right hand side of Eq. (2) scales with the weight

of the ice samples ($Wt_s$), assuming sample-to-sample density variations are negligible. The Eq. (2) can be rearranged to:

$$\frac{COS_{dry}}{(COS_{dry} - COS_{wet})} = C_3 + \frac{C_4}{Wt_s} \qquad (3)$$

     $C_3$ and $C_4$ are constants that can be estimated from the linear regression of the inverse of $Wt_s$ versus the left hand side of equation-3 using data from above 720 m (Fig. A2). We use a Bayesian errors-in-variables linear regression to estimate the solubility correction for the SPC14 wet extraction data due to nonuniform errors, although a least squares linear regression yields similar results (Fig. A3). The errors-in-variables regression is conducted within a hierarchical Bayesian framework using the Stan probabilistic software package (mc-stan.org), in which all uncertain parameters are interpreted probabilistically (Carpenter et al., 2017). Stan uses efficient Hamiltonian Markov Chain Monte Carlo (HMC) sampling. The Bayesian model assumes distributions for the $x$ variable ($1/Wt_s$) using Eq. (4),

$$x \sim normal(x_{true}, \ x_{err}) \qquad (4)$$

which describes the measured $x$ as normally distributed with a mean equal to the true $x$ and standard deviation $x_{err}$. The probability distribution of the linear relationship between $x$ and $y$ (left hand side of equation-3) variables is described by Eq. (5):

$$y \sim normal(b + a \times x_{true}, y_{err}) \qquad (5)$$


     where parameters $b$ and $a$ are the unknown intercept ($C_3$) and slope ($C_4$) of the linear relationship. We do not impose any *a priori* limits on the parameters $a$ and $b$. The true $x$ parameter is required to be positive because sample weight cannot be less than zero. The posterior probability distributions are calculated via the Stan code (supplementary code) as executed in cmdStan 2.29 (https://github.com/stan-dev/cmdstan/releases). The HMC code uses 4 chains, 2000 iterations each for a total of

8000 evaluations of the posterior probability. The HMC diagnostic split $\hat{R}$ achieved $\hat{R} < 1.01$ for the four chains, indicating convergence.

     Once $C_3$ and $C_4$ are known, the solubility correction for wet extraction measurements can be estimated by inserting the $COS_{wet}$ and $Wt_s$ data into equation-3 to calculate the solubility corrected COS mixing ratio represented by $COS_{dry}$ in equation-

3. The calculations are carried out within the generated quantities block of the Stan code (supplementary code) using the full probability distributions of $C_3$ and $C_4$, and the uncertainty distributions of the $x$ and $y$ data. This allows for the propagation of all measurement and parametric uncertainty to the corrected wet extraction COS record. The average solubility correction ($COS_{dry}/COS_{wet}$) is a factor of 1.20 for samples shallower than 720 m and 1.23 for samples deeper than 720 m (Fig. A3).

## 2.3 Calculation of ssNa and nssCa

We use established methods to infer the source contributions of $Na^+$ and $Ca^{2+}$ ions for the SPC14 ice core (Winski et al., 2021). We partition $Na^+$ and $Ca^{2+}$ into sources from marine and crustal origin based on the respective $Ca^{2+}/Na^+$ mass ratios of 0.038 and 1.78 (Bowen, 1979). Given these ratios and assuming that the sum of marine and terrestrial $Na^+$ and $Ca^{2+}$ equals what is measured in the laboratory, we use:

$$ssNa^+ = \frac{Na^+ - \frac{Ca^{2+}}{R_t}}{1 - \frac{R_m}{R_t}} \tag{6}$$

$$nssCa^{2+} = \frac{Ca^{2+} - R_m Na^+}{1 - \frac{R_m}{R_t}} \tag{7}$$

where $R_m$ = 0.038, $R_t$ = 1.78, $nssCa^{2+}$ is terrestrial non-sea salt $Ca^{2+}$ and $ssNa^+$ is sea salt $Na^+$. ssNa and nssCa are commonly used as reliable proxies for ice core impurities of terrestrial and marine origin (Fischer et al., 2007).

## 2.4 Bayesian methods for errors-in-variables regressions of ssNa versus COS

We use errors-in-variables regression analyses between sea salt Na (ssNa) and COS to quantify the excess COS amount, correct the measurements, and recover the underlying atmospheric record. All errors-in-variables linear regression analyses are conducted within the same hierarchical Bayesian framework as the solubility corrections, using the Stan probabilistic software package. The Bayesian model assumes distributions for the $x$ variable based on equation-4. The probability distribution incorporating the linear relationship between x (ssNa) and y (COS) variables is characterized by Eq. (8):

$$y \sim normal(b + a \times x_{true}, \ \alpha \times y_{err}) \tag{8}$$

which only differs from Eq. (5) in that it includes $\alpha$ as a third parameter scaling the y-variable errors as needed depending on various analysis scenarios used in the ssNa vs. COS regressions (Table 2). The error estimates for the COS and ssNa measurements (section 2.5) are directly incorporated in the errors-in-variables regressions as they substitute for $y_{err}$ and $x_{err}$ in equations 4 and 8, allowing us to propagate the uncertainty in the measurements to the slope of the relationship between ssNa-COS. Various data treatment decisions made during the data analysis may act as additional sources of uncertainty. To address this concern, we use different analysis scenarios, four for the glacial period (G1 through G4 in Table 2) and two for the Holocene (H1 and H2 in Table 2), making different data treatment decisions. The factors considered in construction of the

different analysis scenarios include 1) different smoothing methods for COS and ssNa to minimize the impact of uncorrelated high frequency variability in ssNa and COS, 2) different approaches to estimating the measurement uncertainties, including scaling them up, therefore determining the sensitivity of results to the uncertainty in the uncertainty estimates, and 3) conducting simultaneous same-depth and same-age regressions in G4 and H2 to account for the underlying atmospheric COS variability while quantifying the ssNa correction, which should yield a more realistic range for atmospheric COS variability. The corrections are conducted using the full slope probability distribution functions (PDF) for all analysis scenarios and the results are presented in section 3.1.

We do not impose any *a priori* limits on the *a* and *b* parameters. The true x ($x_{true}$) parameter is required to be positive in the all regressions because ssNa cannot be less than zero. In three of the analysis scenarios (G1, G2, H1), the $\alpha$ parameter is constrained to a constant value 1, meaning the estimated y-variable uncertainties were not scaled in the Bayesian inference. The G3 regression includes an unconstrained $\alpha$. The G4 and H2 regressions include simultaneous regressions of the same-depth (co-registered quantities from the same depth in the ice core) and the same-age (quantities from the same ice and gas ages) relationships between ssNa and COS as well as the unconstrained $\alpha$ (Table 2). Inclusion of the $\alpha$ parameter allows evaluation of the sensitivity of the correction slope to possible bias of COS measurement errors. The simultaneous regression algorithm includes two instances of the Eq. (4) and two interdependent equations replace Eq. (8), characterizing the linear relationship for the same-depth and the same-age analyses:

$$y - (a_2 \times x_2) \sim normal(b_1 + a_1 \times x_1, \ \alpha \times y_{err}) \tag{9}$$
$$y - (a_1 \times x_1) \sim normal(b_2 + a_2 \times x_2, \ \alpha \times y_{err}) \tag{10}$$

The simultaneous regression includes 6 unknowns: $x_1$ and $x_2$ represent the true x from Eq. (4) for the same-depth and same-age regressions; $a_1$, $a_2$, $b_1$, and $b_2$ are slopes and intercepts for the same-depth and same-age regressions. The model accounts for the relationships in both spaces by estimating the parameters simultaneously. The simultaneous regression algorithm is the most complex Stan code used in our analyses (supplementary code). Simplified versions were used as needed for the different analysis scenarios described in Table 2. The regressions were conducted with cmdStan 2.29 (https://github.com/stan-dev/cmdstan/releases), using 4 chains, 2000 iterations each for a total of 8000 evaluations of the posterior probability. The HMC diagnostic split $\hat{R}$ (a measure of convergence between and within chains) of the slope posteriors for the G1, G2, G3, G4, H1, H2 scenarios achieved $\hat{R} \leq 1.001$ with respective effective sample sizes ($N_{eff}$) of 4900, 2500, 7100, 3900, 3900, 4700. The convergence criteria are met when $\hat{R} < 1.01$ at $N_{eff} > 400$ (Vehtari et al., 2021). The same diagnostics for all other Stan-based regressions also indicate convergence.

All slope PDFs shown in the manuscript display the posteriors of the slope parameter from the Bayesian inference. We also estimate a probability range for the corrected COS records for all regressions using the "generated quantities" block of the Stan code (supplementary code), accounting for the full range of uncertainties that arise from the slope PDFs and the x-

variable ssNa. The $2\sigma$ uncertainty ranges shown in the Figs. 4b and 5 represent ±2 stdev of the posterior probability distributions.

## 2.5 COS and ssNa errors for the Bayesian linear regression analyses

There are two primary factors that contribute to the error estimates of the individual COS measurements (Aydin et al., 2007). The first one is the uncertainty that arises from the reproducibility of calibration curves. A typical COS calibration curve for the GC-MS system is more uncertain at the low end of its dynamic range. This results in larger relative errors for lower COS mixing ratios and for smaller ice samples that yield less air for the analysis. The second factor is the background COS that is inherent to the extraction vessels and vacuum lines used for the ice core air extractions. The background COS is

characterized by clean $N_2$ analyses through the ice core extraction system, which is subsequently subtracted from the ice core sample signal. The variance in the background COS level introduces a larger relative uncertainty to smaller samples and lower mixing ratios much like the calibration curve uncertainty. As a result, the reported COS measurements have non-uniform errors.

    ssNa was measured continuously in the SPC14 ice core, leading to much higher data density than the COS measurements

(Winski et al., 2021). We use a moving standard error of measurement means that fall within a time window (typically 100 y or higher and lower depending on the analysis scenario) to calculate the uncertainty in the ssNa concentration at the corresponding depth range in the ice core. The standard errors are calculated from unbiased (divided by N-1) standard deviations. There are on average 82 ssNa measurements per 100-y averaging window below 1200 m, which is the depth range corresponding to the glacial period dry extraction measurements, which constitute the basis for excess COS corrections.

**3 Results**

    COS was measured over the length of the SPC14 ice core with the shallowest samples from 129 m and the deepest one from 1749.3 m depth (Fig. 1a). Out of a total of 574 measurements, 425 samples were analyzed by dry extraction and 149 by wet extraction. A solubility correction (~20% on average) is applied to all wet COS measurements (Fig. A3). No hydrolysis loss correction is applied to any of the SPC14 COS measurements. The solubility-corrected wet measurements exceed the 35

dry measurements in the 800-900 m range (Fig. 1a), indicating a depletion in these dry measurements due to the onset of the BCTZ. The onset of BCTZ around 800 m is also evident from the large drop in extracted gas amount, reflecting the associated decrease in gas extraction efficiency during dry extraction (Fig. A1a). These 35 measurements (0.6% of the data set) are excluded from data analyses.

    The remaining measurements (n=539) exhibit higher variability below 1035 m (16 ky in gas age), displaying a

propensity for rapid positive excursions (spikes) of 100-150 ppt (Fig. 1a). Consequently, the distribution of the measurements prior to 16 ky is broad and skewed compared with the measurements from periods after 16 ky (Figs. 2a, b). Both dry and wet extraction measurements prior to 16 ky display similarly skewed distributions, implying that the spikes do not result from

inefficient gas extraction or another experimental artifact impacting the clathrate ice measurements. The atmospheric lifetime of COS (2-3 yrs) and the spectral width of the firn smoothing at the South Pole (50-80 y; Epifanio et al., 2020) are both shorter than the average resolution of the dry glacial period measurements (160 y), therefore we cannot rule out the possibility that these spikes represent atmospheric variability. However, 100-150 ppt spikes imply 30-50% changes in COS biogeochemistry over 100-y time scales, which is similar to the increase in atmospheric COS during the 20[th] century caused by anthropogenic activities (Aydin et al., 2020). It seems unlikely that the COS mixing ratio in the glacial period atmosphere varied abruptly at the magnitude and frequency of these spikes. There is a prominent peak in 1114-1144 m depth range characterized by four wet measurements which are considerably higher than the adjacent data, hence more likely to be indicative of an atmospheric peak than the shorter lived spikes (Fig. 1a).

There are two other ice core COS records that extend back to the last glacial period. They are from the Taylor Dome (TD) and the West Antarctic Ice Sheet Divide (WD), Antarctica (Fig. 1b). Both of these sites are warmer than the South Pole, therefore the TD and WD measurements require a correction for temperature-dependent hydrolysis loss (Aydin et al., 2014; 2016). Despite this correction, there are discrepancies between the COS records from the different sites. The SPC14 and TD records agree during the Holocene, but the TD data are consistently lower than the SPC14 during 45 – 10 ky. The WD record is in better agreement with the SPC14 record than the TD prior to 20 ky, mostly tracking the lower bound of the SPC14 record, but it trends in the opposite direction from both the SPC14 and the TD during most of the deglaciation into the early Holocene from 15 – 11 ky. The high scatter evident in the WD record from 11 – 8 ky and the following deep trough around 6 ky occurs within the BCTZ at the WD site, which also impacts other dry extraction-based trace gas measurements from that site such as the $CO_2$ record (Marcott et al., 2014). Spikes similar to the ones apparent in the glacial period SPC14 record are also evident in the other two ice cores, near 36 ky and 42 ky at the TD and 50 ky and 37 ky at the WD sites.

The gas chronologies of the SPC14 and the WD ice cores are synchronized by methane measurements to within 100 y (Epifanio et al., 2020), therefore the diverging trends between the SPC14 and WD records from 15 – 11 ky cannot be attributed to chronology uncertainties. The TD ice core has an independent gas chronology with documented problems over various depth ranges (Ahn and Brook, 2007; Monnin et al., 2004), but the persistent low bias apparent in the TD record prior to 10 ky with respect to SPC14 cannot be attributed to chronology uncertainties. Inaccuracies in the kinetic parameters of hydrolysis loss or the temperature histories in the temperature histories can result in the TD record from 45 – 31 ky to be biased low; however, this was deemed unlikely (Aydin et al., 2016).

Alternatively, the discrepancies between the records could be due to production in the ice sheet resulting in significant amounts of excess COS in glacial period ice; this possibility was not considered by Aydin et al. (2016). Impurity concentrations are higher in glacial period ice at all Antarctic sites. The timing of the opposing trends in measurements from different ice cores supports the possibility of COS production linked to high impurity concentrations. For example, in the SPC14 record COS levels decline down-core from 11 to 13 ky, then increase from 13 to 16 ky (Fig. 1b). This reversal occurs at a depth where ice impurity concentrations are increasing steeply (Fig. 1c). The presence of impurity driven production would mean that the previous inferences on GPP variability during the deglaciation by Aydin et al. (2016) are not valid.

We assess the potential for COS production in the SPC14 ice core using ssNa and nssCa concentrations from the COS measurement depths (same-depth analysis) as proxies for ice impurities of oceanic and terrestrial origin. We initially use only the dry COS measurements from the 1200 – 1751 m depth range (23 – 52 ky). These samples were about 20 cm long. The mean annual layer thickness in this depth range is 2 cm, so each sample averages 10 years of ice chemistry. Linear regressions were conducted between measured COS mixing ratio and 10-y, 100-y, and 300-y averaged impurities (ssNa and nssCa) to determine whether marine or terrestrial impurities correlate with COS (Fig. A4). We primarily rely on Bayesian errors-in-variables linear regressions (section 2.4) because both the $x$ and $y$ variables have non-uniform errors (section 2.5). Standard least-squares linear regressions are also conducted for comparison. The time averaged impurities are centered around the mid depths of the COS samples regardless of the averaging window; the sensitivity of results to this choice is tested in section 3.2.

We find statistically significant slopes between COS and both ssNa and nssCa from the same depth regardless of the averaging window for the ice impurities, indicating some level of COS production from ice impurities (Figs. A4). The slope is more significant with ssNa than nssCa for all averaging windows; we interpret this to be indicative of COS production being linked to marine sourced impurities because ssNa and nssCa are correlated in the SPC14 glacial period ice, although the relationship is non-linear, resulting in the weaker COS-nssCa correlation. 10-y average ssNa yields a significant slope (p=0.003) with COS, but the lowest $R^2$ (0.04) (Fig. A4). The significance of the slope (p=0.000, full PDF for errors-in-variables is displayed in Fig. 3a) and $R^2$ (0.08) increase when the ssNa averaging window is increased to 100-y and slightly decline at 300-y. Note that the COS spikes, which do not correlate with ssNa variability at the same frequencies, contribute to the relatively low $R^2$ of these relationships. When the COS and the ssNa records are both smoothed to characterize the linear relationship over longer time scales, the $R^2$ values increase (Fig. A5).

In subsequent analysis, we find significant correlations between COS and ssNa in the enclosing ice above 800 m (Holocene ice), providing compelling evidence for COS production regardless of the average ssNa concentrations across glacial to interglacial periods (Figs. 2c-d, A5, A7). The fact that the same-depth relationship between ssNa and COS is stronger when ssNa is averaged over 100 years or more implies that the higher frequency variance in COS is not explained by ssNa variance at similar frequencies (Figs. 1b, A4, A5). As noted above, the presence of spikes is a contributing factor to the low $R^2$ of the correlations despite the high significance and we later show that the spikes in the COS signal do not significantly impact the quantification of the relationship with ssNa by smoothing both the COS and ssNa signals in different analysis scenarios (e.g. G2 described in section 3.1). We find no evidence in the soluble ions and dust records that correlate with the spikes evident in the pre-16 ky COS measurements and the SPC14 record does not display a downcore increasing trend or a downcore increase in the magnitude of the COS spikes. These observations suggest ssNa is a proxy for a production process confined primarily to the firn column. If some level of additional production happens in the ice sheet after lock-in or in the lab during extraction, which are the types of processes that can result in sharp spikes, the high-frequency variations in bulk measures of marine (ssNa) or terrestrial (nssCa) aerosols are not good proxies for it.

We considered the possibility of the correlation with ssNa arising from a climate-driven relationship between ssNa and COS that somehow influences the same-depth regressions; however, during the glacial period the difference between ice age

and gas age (delta-age) for the SPC14 ice core is large and temporally variable (Fig. A1b), implying this is unlikely. Indeed, same-age correlation analyses display a negative correlation during the glacial period, ruling out the possibility of a climate driven relationship as the explanation for the positive same-depth relationships (Figs. 3c, A6). The same-age analysis reveals a positive relationship during the Holocene, but this does not negate the same-depth relationship (Figs. 3d, A7). Rather,

simultaneous regressions of COS versus same-depth and same-age ssNa (section 2.4) confirm that two independent relationships exist between ssNa and COS, the same-depth one driven by production, and the same-age one that can be interpreted as a climate-mediated variability in the ice core COS record.

**3.1 Results of different analysis scenarios: Inferred atmospheric COS variability after the ssNa correction**

The results from all of the analyses scenarios yield similar estimates of atmospheric COS variability after correction for

the presence of excess COS in the ice core measurements (Fig. 4b). In G1, the errors-in-variables slope of the 100-y ssNa vs. COS same-depth regression is used to correct the COS record (Figs. 2-4, A5a-c). This scenario does not include any smoothing of the COS measurements, which leads to a relatively low $R^2$ partially because high-frequency variability in COS (the spikes) do not correlate with ssNa. In G2 (Fig. A5d-f), we smooth the COS measurements with a 7-point moving average, which approximates to an 1143-y averaging window, and use the 7-point moving standard error of the measurements themselves as

the $y$-variable error, in place of the original measurement errors. The ssNa measurements are also smoothed to 1100 years to match COS. Collectively, these choices lower the influence of the COS spikes from the regression via smoothing and introduce higher uncertainty to the $y$-variable COS where there is a higher occurrence of COS spikes. The $R^2$ for G2 (0.22) is significantly higher while the mean slope estimate increases marginally, indicating the same-depth ssNa-COS relationship is driven primarily by millennial scale variability and is not influenced by the higher frequency COS spikes.

The analysis scenario G3 (Fig. A5g-i) differs from G2 primarily in that we reduce the weight of COS spikes on the regression by smoothing the COS record by a 7-point moving median. The treatment of $y$-variable errors also changes. We use a 7-point root mean square of the original COS measurement errors and introduce a constant multiplier ($\alpha$ in Eqn. (8)) to allow the Bayesian inference to scale the $y$-errors as needed (section 2.4). The G3 scenario demonstrates the impact of minimizing the influence of COS measurement uncertainties on the errors-in-variables analysis as the $y$-variable errors in G3 are more

uniform by design and are scaled up by more than a factor of three via the $\alpha$ parameter (Fig. A5l), resulting in a closer slope estimate to the least squares linear regression of the data used in the G3 scenario (compare slopes in Fig. A5g versus G3 in A5i).

In G4 (Fig. A5j-l, A6f-i), we introduce a simultaneous same-age regression of COS with ssNa. As noted earlier, the same-age correlation between COS and ssNa is interpreted as a climate driven relationship. The delta-age is 1500-2700 years

over the 1200 – 1751 m depth range (Fig. A1b) corresponding to 30-54 m of ice for an average annual layer thickness of 2 cm, implying only multi-millennial scale climate trends would have an impact on accurate quantification of the same-depth slope. We also conduct an independent same-age only regression of measured COS vs. 100-y smooth ssNa (Fig. A6a-c, same-age regression for G1). The same-age regression slopes for G1 and G4 have mostly overlapping PDFs and similar means (Fig. 3c),

revealing a climate-driven negative correlation between ssNa and COS over multi-millennial time scales which impacts accurate quantification of the slope of the same-depth relationship (Fig. A5i). Detrending COS with the same-age ssNa-COS relationship increases the $R^2$ of the same-depth regression to 33% (Fig. 2c, Fig. A5j). The G4 results are a more accurate estimate of the same-depth slope than G3 as it implicitly accounts for the influence of the climate driven variance in the record while quantifying the same-depth relationship between ssNa and COS.

We conducted two similar data analysis scenarios (H1 and H2) for the dry extraction measurements from above 800 m. In H1 (Fig. A7a-c, g-i), the same-depth and same-age correlations are identified independently using COS measurements (49-y resolution on average) and 100-y smooth ssNa. In H2 (Fig. A7d-f, j-l), we apply a 7-point smoothing to the COS record (350 y smoothing window on average) and conduct simultaneous same-depth and same-age regressions vs. 300-y smooth ssNa. We find significant positive correlations between ssNa and COS from the same depth and from the same age. Quantification of the same-age relationship with the simultaneous regression does not significantly change the mean slope of the same-depth relationship but results in a narrower probability distribution, hence increases its significance (Fig. A7c,f). The results of the different scenarios display the most likely range of atmospheric COS variability over the last 52 ky with the 2σ uncertainty ranges representing ±2 stdev of the posterior probability distributions from the different scenarios (Fig. 4b).

Collectively, the different analysis scenarios yield comparable estimates for average COS levels over long time scales and all imply lower atmospheric COS during the last glacial period (Fig. 4b). The inferred atmospheric record averages about 150 ppt from 52.5 ky to 25 ky, which is more than 100 ppt lower than the Holocene mean (Fig. 3f). During the late Last Glacial Maximum (LGM) around 18 ky, the probable range for atmospheric COS falls below 100 ppt, after which atmospheric levels rise through the deglaciation to reach 250-300 ppt during the Holocene. Note that at 18 ky, atmospheric COS is more likely to be less than 50 ppt than higher 100 ppt, and 150 ppt is the upper limit.

We conducted additional analyses to test the sensitivity of results from the different scenarios to 1) minor depth offsets to the ssNa data used in the regressions and 2) elimination of some fraction of the data from the regressions. We find that the best correlations are achieved when COS is regressed versus ssNa from the same depth and that the results are repeatable at lower data densities but significance gets progressively worse with less data (Appendix B). Next, we tested the robustness of the same-age relationship between ssNa and COS with respect to the applied same-depth correction and found that the temporal relationship with ssNa during the glacial period is likely inherent to the COS data set and do not result from the applied same-depth ssNa correction (Appendix C). Finally, we also tested whether there is any explicit statistical evidence that the increase in the inferred atmospheric COS record concurrent with the deglaciation during 19-10 ky is climate driven as opposed resulting solely from the applied ssNa-correction. We find that the deglacial increase in atmospheric is more similar to the deglacial climate signal than the inverse of the applied ssNa correction (Appendix D).

### 3.2 Comparison of the SPC14 record with records from other Antarctic ice cores

Comparison of the SPC14 record to previous ice core measurements is challenging. There is no other COS data set that does not require a hydrolysis loss correction (i.e. no data from another very cold site) or at a comparable resolution to the

SPC14 record from the glacial period that would allow an investigation of impurity driven COS production impacting the measurements. We presume that some COS production occurs at every site and this is the reason why the WD and TD records trend in different directions during the late deglaciation even after the hydrolysis loss correction. Subsections of the COS measurements from the TD and WD sites do not require large corrections for the hydrolysis loss due to relatively colder temperature histories: during the glacial period at WD and during mid-to-late Holocene at TD. We make use of this fact to conduct a preliminary analysis to determine whether a same-depth relationship exists between ssNa and COS at these sites during these periods.

The same-depth relationships between ssNa and COS in TD during the Holocene and WD during the glacial period and deglaciation are similar to the SPC14 observations (Fig. A11). Given the impact of lower data density on regressions (section 3.2), instead of determining site specific correction slopes, we correct the TD and WD measurements using the G4 correction slope estimate from the SPC14 ice core (Fig. 5). The COS production occurs primarily in the firn whereas hydrolysis loss is a slow process occurring over multi-millennial time scales (Aydin et al., 2014; 2016). Therefore the ssNa correction is applied to the loss-corrected COS records. This comparison is considered preliminary and evaluated qualitatively given that the general applicability of the same-depth ssNa-COS regression slopes has not been tested.

After the ssNa correction, the Taylor Dome record agrees better with the SPC14 record during the glacial period from 45 – 20 ky and also suggests a steep increase during the deglaciation similar to the SPC14 record (Fig. 5). The timing of the deglacial COS increase is somewhat different for the Taylor Dome record versus the SPC14 record. The discrepancies could result from the use of constant and SPC14 specific ssNa correction parameters, but there may be other contributing factors. The full range of uncertainty in the Taylor Dome loss correction arising from the uncertainties in the temperature histories have not been quantified (Aydin et al., 2014; 2016). Further, the Taylor Dome chronology during the LGM and the deglaciation is susceptible to larger uncertainties than the SPC14 ice core, including known errors during the early Holocene because the current Taylor Dome gas chronology is tied to the outdated EDC-1 chronology (Monnin et al., 2004). Chronology errors impact the hydrolysis loss correction too as this correction is both temperature and time dependent.

The ssNa-corrected WDC-06A record agrees better with the SPC14 record during the glacial period and through 14 ky, capturing the first few thousand years of the COS rise during the glacial/interglacial transition (Fig. 5). However, the WDC-06A does not show any indication of a peak near 19 ky. The WDC-06A record also displays a deep trough around 13 ky. The relative age uncertainty between the SPC14 and WD ice cores is 100 y or less for most of the core (Epifanio et al., 2020). At least part of the discrepancy between the three ice cores during the late deglaciation might be due to the fact that the SPC14 ssNa correction parameters are not broadly applicable to the Taylor Dome and WAIS Divide ice cores during these periods. The other contributing factor could be the unquantified uncertainties in the hydrolysis loss corrections applied to the WDC-06A and Taylor Dome records.

## 4 Discussion

### 4.1 COS production in the ice sheet

Production of trace gases can occur both in the firn and in deeper ice below bubble lock-in presumably from ice impurities (Butler et al., 1999; Fain et al., 2014; Mühl et al., 2023). If the excess gas amount is comparable to the mean atmospheric levels, a correlation between the gas and the ice impurities would start to emerge. If production starts immediately below bubble close-off and stops after a relatively short period, the excess gas would correlate with the impurity concentrations measured over the exact same depth range as the gas sample. This scenario is conceptually unrealistic because there is no

obvious reason why there would not be a firn component to any production process that takes place in the ice sheet, with the only exception being production in very deep, warm, and wet ice possibly with incorporated bedrock material. It could be reasonable to ignore the firn component if the production continues over much longer time horizons than the firnification time scale. If this is case, the continual accumulation of excess gas over time should introduce a long term increasing trend to the gas record that would be absent from the impurity record underlying the production. In contrast, if the trace gas production is

short-lived, production in the firn could be the primary component, in which case the excess gas is expected to correlate with the impurities but over a larger depth range than the exact length of the ice core samples because of vertical mixing of gases in the firn.

      The relationship between the SPC14 COS measurements and ssNa favors a process primarily taking place in the firn because there is no long term trend in excess COS (i.e. production is a relatively short-lived process) and the correlation with

ssNa is driven by longer time periods than the age range that corresponds to the length of our samples. In theory, it is also possible gas production can happen in the laboratory during the extraction of the ice core air. We deem this unlikely because production during extraction should also result in a correlation with impurities measured over the exact same depth range as our samples. Further, the agreement between COS measurements with two different extraction methods implies excess COS is generated in the ice sheet. The production process may be short-lived because reaction substrates run out over time, lowering

the probability of reactions as the ice ages. Further, solid and dissolved phase ice impurities, e.g. sea salt aerosols and trace metals that can act as reaction substrates and catalysts for production of COS, can migrate to different locations via post-depositional processes that occur in shallow ice (Stoll et al., 2023), also lowering the probability of complex reactions deeper in the ice sheet.

      To demonstrate the viability of the proposed production process as an explanation for our observations, COS production

in the firn is simulated within an advective-diffusive model of the South Pole firn (Aydin et al., 2020). The model tracks a layer of impurity (e.g. ssNa) that induces gas production (e.g. COS) at a fixed rate (1 ppt yr$^{-1}$) through its progression in the firn from the surface to the bottom (Fig. 6). In this framework, the excess gas peak at any given depth in the firn is always broader than the production peak in both directions (up and down the firn) because as soon as the gas is produced, it starts to mix vertically. The model also demonstrates that the excess gas produced in the lock-in zone (below 100 m at the South Pole)

accumulates with much higher efficiency, resulting in progressively sharper peaks with depth. Another important implication

is that the $R^2$ statistic between the excess gas and the impurity concentration at the same depth is considerably less than 1 even in the absence of any underlying atmospheric variability or measurement uncertainty (Fig. 6).

In the absence of direct empirical evidence, we can only speculate on viable mechanisms for the COS production in the firn. ssNa approximates marine aerosol deposition to the ice sheet (Winski et al., 2021). It follows that the COS production mechanisms might be similar to the dark production mechanisms in the ocean. We focus on dark production because the photochemically active zone in an ice sheet is largely constrained to the top meter of the firn (Grannas et al., 2007; King and Simpson, 2001) and our model simulation shows only a tiny fraction of the gases produced in shallow firn can be preserved in the ice sheet (Fig. 6). There is not a well-established mechanistic explanation for dark production of COS in the ocean (Cutter et al. 2004; Von Hobe et al., 2001), but recent studies favor chemical mechanisms over biological ones (Lennartz et al., 2017; 2019). The commonly postulated abiotic process involves reactions between carbonyl groups and thiyl radicals derived from organic sulfur (Flöck et al., 1997; Modiri Gharehveran and Shah, 2018; Lennartz et al., 2017; 2019; Pos et al., 1998; Zepp and Andreae, 1994). Organic salts have been used as carbonyl donors for COS production in laboratory settings (Pos et al., 1998). Sea salt aerosols host a variety of chemical reactions during atmospheric aging, including reactions between NaCl and organic acids leading to production of organic salts of Na (Laskin et al., 2012). This could explain why ssNa works as a proxy for excess COS. Abiotic COS production also requires an organic sulfur source. Antarctica is far from continental land masses, suggesting marine aerosols could also be the primary source of organic sulfur. Reactions between organic salts and sulfur can be catalyzed in the dark by transition metals (Modiri Gharehveran and Shah, 2018; Lennartz et al., 2019; Pos et al., 1998).

**4.2 Interpretation of the atmospheric COS record**

The ssNa correction is relatively small during the Holocene (Fig. A1c) and implies 10-15% less COS in the preindustrial atmosphere than what was previously inferred from uncorrected measurements (Aydin et al., 2020). Excess COS of this magnitude is difficult to identify with firn measurements because of the uncertainties involved in the firn measurements and inversions. There is a positive same-age correlation between ssNa and COS during the Holocene that arises primarily from the dip centered around 6 ky in both records, and to a lesser extent a subsequent long-term rise followed by a decrease over the most recent 700 years (Fig. A7k). The ssNa variability apparent in the SPC14 ice core during the Holocene tracks winter sea ice extent in the Southern Ocean (Winski et al., 2021). Modern day winter sea ice concentrations around Antarctica have been linked to organic acid emissions from phytoplankton blooms, which also emit DMS (Curran and Jones, 2000; King et al., 2019). Direct COS fluxes are also high in the Southern Ocean, but have not been linked to sea ice (Kettle et al., 2002; Lennartz et al., 2017; 2019). It is possible that atmospheric COS is sensitive to changes in ocean DMS emissions modulated by winter sea ice. Further research is needed on whether COS is co-emitted with DMS along the sea ice margins.

During 52 – 25 ky, the 2σ confidence interval of G4 mostly remains between 100 – 200 ppt averaging about 150 ppt (Fig. 4c), which is lower than the mean levels during the Holocene by about a factor of 2 (Fig. 3f). This difference is quantified directly by the dry measurements and is highly significant. During the LGM minimum delineated by the wet measurements, the probable range of atmospheric COS is less than 100 ppt, implying in a rise by a factor of 3-4 concurrent with the

deglaciation. In section 3.3, we show that the same-age anticorrelation with ssNa is evident even in the regression of measured COS with ssNa (Fig. 3c) and not caused by the same-depth correction. The analysis in section 3.4 provides strong statistical evidence that the temporal evolution of the deglacial COS arises from the variability inherent to the COS measurements. The deep COS minimum at the end of the LGM appears to result from the culmination of a long term trend that starts prior to 30 ky and is climate driven.

Shorter period temporal variability apparent during the glacial period requires additional scrutiny. The methods we use in applying the ssNa correction do not explicitly address the possibility of temporal variability in the ssNa correction slope. Therefore, the actual temporal trends in atmospheric COS may not track the mean of the posteriors for any of the analysis scenarios. However, the reported $\pm 2\sigma$ uncertainty ranges are determined by correcting the COS measurements with a range of slope values drawn from the posterior PDFs (Fig. 3a, b) while also accounting for measurement uncertainties. If the same-depth ssNa-COS relationship varies over time but stays within the range of the slope PDFs, the inferred COS variability should remain within the reported uncertainty ranges, particularly over long time scales. Temporal variability in the correction slopes that exceed the posterior ranges of the different scenarios cannot be ruled out; however, such temporal variability could only last for brief periods such that it would not alter the long term mean states described by the slope posteriors. In the remainder of this section, we will primarily focus on the interpretation of the deglacial increase in mean COS levels. We do not offer a biogeochemical explanation for the COS spikes because it is unlikely that they represent atmospheric variations, with the possible exception of the sharp peak around 19 ky.

The 19 ky peak is characterized by four wet COS measurements and coincides in time with a shorter-lived sharp peak in ssNa (Fig. 4c). Given the prominence of spikes in the glacial period COS, we suspect that at least one of the measurements characterizing the 19 ky COS peak, possibly the highest measurement dating older than 20 ky, may be a coincidental, non-atmospheric spike while the other three measurements may characterize an atmospheric excursion of 50-100 ppt that is closer in duration to the peak in ssNa. This may explain why we do not see this feature in the WD COS record. An atmospheric COS excursion of this magnitude would represent a sudden and significant departure from the biogeochemical balance that maintains the low atmospheric COS levels during the LGM. Based solely on the magnitude, it could only be caused by an increase in ocean sulfur gas emissions or a decline in land biosphere uptake since these are the two major natural components of the COS budget (Table 1). This feature warrants further investigation if replicated with high resolution measurements from different ice cores.

The 2-4 fold increase in atmospheric COS across the deglaciation implies equivalent changes in the sources or the removal processes. The atmospheric lifetime of COS is linked to the gross primary productivity (GPP) of land plants, atmospheric carbon dioxide ($CO_2$) levels, and the leaf-scale relative uptake (LRU) of COS versus $CO_2$ (Stimler et al., 2010). $CO_2$ is relevant because the first order loss rate constant for atmospheric COS via plant uptake ($k_{veg}$) is proportional to stomatal conductance, which is inversely proportional to $CO_2$ (Stimler et al., 2010):

$$\frac{1}{\text{lifetime}} = k_{veg} \propto \frac{\text{GPP} \times \text{LRU}}{[CO_2]} \qquad\qquad (11)$$

In equation-11, a change in atmospheric $CO_2$ results in a proportional change in the COS lifetime if GPP and LRU are
525 constant. Atmospheric $CO_2$ increases from 190 ppm to 270 ppm (factor of 1.4) during the deglaciation (Fig. 4c), which implies
an equivalent increase in the COS lifetime.

Of course many large-scale changes in the terrestrial biosphere occur during the deglaciation and it is unlikely that GPP
and LRU remain constant. The warmer interglacial world has less ice sheet cover and more $CO_2$ in the atmosphere, promoting
a general expansion of plant coverage that favors woody plants (C3) over grasslands (C4) and an increase in GPP by 40-50%
(Ciais et al., 2012; Prentice et al., 2011). The effect of increasing GPP is equivalent to but in the opposite direction of the $CO_2$
effect on the COS lifetime. LRU varies with light, humidity, and $CO_2$, but these effects are small for realistic ranges of change
in global average conditions (Sun et al., 2022). LRU is relatively more sensitive to photosynthetic pathways and is higher for
C3 plants than C4 plants (Stimler et al., 2011; Sun et al., 2022), suggesting this effect would dominate the LRU response to
the deglaciation and result in a shorter lifetime during the Holocene. Since the other loss processes are too small (Table 1), we
presume that all or most of the COS rise during the deglaciation is driven by emissions and has to involve all major sources
since the required increase is very large. If the COS lifetime were shorter during the Holocene, the proportional increase in
emissions during the deglaciation would have to be higher than the increase apparent in the COS mixing ratio.

Biomass burning likely increased during the deglaciation (Bock et al., 2017); the anoxic sources would also have to if
their emissions are closer to the upper limit of estimates. Still, oceans are the largest COS source and most of the required
increase must come from the combined emissions of COS, $CS_2$, and DMS (Berry et al., 2013; Glatthor et al., 2015; Kettle et
al., 2002; Kuai et al., 2015; Lennartz et al., 2017; Ma et al., 2021; Whelan et al., 2018). Photochemical production mediated
by chromophoric dissolved organic matter (CDOM) is the primary production mechanism of COS in the surface ocean, with
an additional dark (light-independent) channel (Flöck et al., 1997; Modiri Gharehveran and Shah, 2018; Lennartz et al., 2017;
2019; Von Hobe et al., 2001; Zepp and Andreae 1994). COS is chemically lost to temperature-dependent hydrolysis in the
545 surface ocean and emissions occur primarily in temperate waters and higher latitudes where hydrolysis is slow (Berry et al.,
2013; Kettle et al., 2002; Lennartz et al., 2017; Whelan et al., 2018). Low latitude emissions of COS are predominantly in
coastal waters and upwelling regions where CDOM concentrations are high (Kettle et al., 2002; Lennartz et al., 2017; 2019;
Nelson and Siegel, 2013; Zepp and Andreae, 1994). The primary $CS_2$ production mechanisms are photochemical much like
COS but possibly involve a different class of organosulfurs that correlate with dissolved organic sulfur in the water; dark
production and production by phytoplankton are also possible. $CS_2$ emissions occur primarily in the low latitudes as chemical
loss rates are negligibly slow even in warm waters (Kettle et al., 2002; Modiri Gharehveran and Shah, 2018; Lennartz et al.,
2017; 2019; 2020; Xie and Moore, 1999). DMS is produced from dimethylsulphoniopropionate (DMSP), a phytoplankton
product, and water column losses are due to microbial metabolism and photosensitized oxidation (Andreae, 1990; Charlson et
al., 1987; Vogt and Liss, 2009; Wang et al., 2018a). Ocean DMS concentrations are generally high where COS emissions are

high, but area integrated emissions peak at the lower latitudes and mid-latitude southern hemisphere where $CS_2$ emissions occur (Lana et al., 2011). DMS is also a weak precursor of COS in seawater (Modiri Gharehveran and Shah, 2018; Zepp and Andreae, 1994) and emissions of all three sulfur gases can be co-located, with COS, $CS_2$, DMS emission enhancements in nutrient-rich coastal seas and upwelling regions (Andreae, 1990; Charlson et al., 1987; Ksionzek et al., 2016; Lana et al., 2011; Lennartz et al., 2017; 2019; Wang et la., 2018a; Whelan et al., 2018; Xie and Moore, 1999). The mechanistic understanding of COS production from atmospheric oxidation of DMS is evolving (Khan et al., 2021; Jernigan et al., 2022) and the upper limit for this indirect source is uncertain (Table 1).

The similar timing of the atmospheric COS and $CO_2$ increases during the deglaciation suggests that COS is responding to global environmental changes driven by climate. Expected temperature and pH changes during the deglaciation may result in weak enhancements in the net oceanic emissions of sulfur gases via impacts on solubility and hydrolysis but these effects are not on the same order as the apparent COS increase during the deglaciation (Appendix E). High magnitude emission changes can be caused by increases in the surface ocean concentrations of dissolved organosulfurs and CDOM, which are primarily byproducts of biogenic productivity and microbial degradation of the biomass (Ksionzek et al., 2016; Nelson and Siegel, 2013). Ocean dynamics, sea level rise, and sea ice extent are physical environment variables that can influence biogenic processes and  alter ocean sulfur gas emissions.

Much of the excess $CO_2$ in the interglacial atmosphere is attributed to increased upwelling of old carbon from the deep Southern Ocean that also elevate macro nutrient and phytoplankton levels around Antarctica (Anderson et al., 2009; Burke et al., 2012; Freeman et al., 2016; Jaccard et al., 2013). Deeper waters are nutrient rich because of remineralization, which also results in higher CDOM in the ocean interior (Nelson and Siegel, 2013). The oceanic mechanisms responsible for the deglacial increase in atmospheric $CO_2$ can enhance DMS emissions via increased phytoplankton biomass and COS emissions by supplying more organic sulfur and CDOM in the Southern Ocean. Decreasing sea ice extent in the Arctic and around Antarctica can further enhance DMS and COS emissions at high latitudes due to increased light availability and air-sea gas exchange (Vogt and Liss, 2009; Wang et al., 2018a,b). Concurrent enhancement of $CS_2$ emissions necessitates activation of processes in the low latitude oceans such as the proposed productivity increase in the equatorial Pacific induced by upwelling of nutrient rich waters (Costa et al., 2017; Winckler et al., 2016), which should result in a co-enhancement of DMS emissions. Additionally, the coastal emissions of COS, $CS_2$, and DMS can increase as a response to the deglacial sea-level rise and the associated expansion of shelves and coastal seas coupled with increased riverine output of organic matter (Jennerjahn, 2012; Lerman et al., 2011; Peltier and Fairbanks, 2006).

There is paleoclimate evidence for changes in upwelling regimes in both high and low latitudes as well as changes in process in coastal regions and on continental shelves that can impact ocean emissions of COS, $CS_2$, and DMS during the deglaciation. We cannot quantify how much of the ocean COS source increase results from DMS versus COS and $CS_2$ because of the complexities of ocean production mechanisms of sulfur gases and the uncertainties in their contribution to the atmospheric COS budget. We suggest that emissions of all three gases are likely to have increased because the geographic distribution of the relatively better quantified ocean sources of atmospheric COS, i.e. direct COS and indirect $CS_2$ emissions,

cover most of the global ocean. The net open ocean emissions of COS and $CS_2$ are geographically decoupled (Lennartz et al., 2020) but open ocean DMS emissions overlap with net emissions of both gases, and all three gases are emitted at coastal upwelling zones. All ocean sulfur gas emissions ultimately stem from organic life in the ocean and biogenically driven enhancements of COS and $CS_2$ production in the surface ocean are unlikely to occur without accompanying changes in DMS production in the same direction. Further, when emissions of all three gases increase, relatively smaller increases are required for emissions of each gas to achieve the same overall change in atmospheric COS. If only COS and DMS or $CS_2$ and DMS increased, i.e. if the emission changes were confined to either the high or the low latitudes, much larger changes in gas emissions would be required within that region to make up for the missing increase from the other region. If the atmospheric COS budget evolves in such a direction that the ocean COS sources are found to be more regional in nature rather than being globally distributed, the deglacial increase in atmospheric COS would consequently imply changes in only those regions of the ocean.

There is independent evidence for a regional enhancement of ocean sulfur gas emissions during the deglacial cycles. A recent interpretation of ice core sulfate data suggests large increases in ocean DMS emissions around Antarctica concurrent with the last 7 deglaciations including a doubling during the last deglaciation (Goto-Azuma et al., 2019), which is consistent with our assessment but in disagreement with the earlier interpretations of the ice core sulfur records (Legrand et al., 1991; Wolff et al., 2006). Unlike the ice core sulfate records, atmospheric COS levels over Antarctica are sensitive to global emissions of DMS due to the longer atmospheric lifetime of COS versus sulfate. DMS emissions in the Southern Ocean alone constitute about 10% of global DMS emissions (Lana et al., 2011), therefore are not nearly sufficient to account for the deglacial COS increase.

The climate impacts of COS and DMS are in the same direction and additive, although DMS is by far the more dominant of the two. The radiative forcing expected from a 150 ppt increase in COS is within -0.004 $Wm^{-2}$ to -0.045 $Wm^{-2}$, with the large uncertainty arising from model discrepancies and uncertainties in the effects of different emission geometries (Brühl et al., 2012; Quaglia et al., 2022; Appendix F). The global net radiative impact of DMS emissions today is estimated at -1.8 to -2 $Wm^{-2}$, with average summer time values over the southern oceans exceeding -9 $Wm^{-2}$ (Thomas et al., 2010; Wang et al., 2018a, b). During the last deglaciation, the total greenhouse gas ($CO_2+CH_4+N_2O$) forcing was +2.5 to +3.0 $Wm^{-2}$ (Kohler et al., 2017). A doubling of global DMS emissions during the deglaciation could have resulted in global radiative forcing of -1 $Wm^{-2}$, offsetting about a third of the concurrent greenhouse gas forcing, with large regional impacts over the Southern Ocean.

## 5. Conclusions

COS measurements from the SPC14 ice core provide evidence for production in the ice sheet that primarily occurs in the firn. The resulting excess COS is a temporally smooth signal, significantly elevating the baseline of measurements from glacial period ice with high impurity concentrations. The measurements also display large spikes (high frequency variability), which visually dominate the record from the glacial period ice, although their impact on interpretation of the record is not

nearly as consequential as the excess COS resulting from production in the firn. The results highlight the importance of evaluating the atmospheric fidelity of trace gas measurements with measurements from glacial period ice with high impurity content. Firn measurements from different sites may not reveal production in the firn if the excess amount under present-day conditions is small compared with the atmospheric background and should never be considered a good analogue for possible

production in the firn during the glacial periods.

We identify ssNa as a proxy for the excess COS in the SPC14 ice core and implement a proxy-based correction method to recover an atmospheric record. It is possible the same approach can be extended to other ice cores, but high resolution measurements from other sites are necessary to investigate the general applicability of this approach. The atmospheric record inferred from the SPC14 measurements imply higher COS levels during the Holocene than the last glacial period with a large

increase concurrent with the last deglaciation. The deglacial COS rise results from an overall strengthening of atmospheric COS sources, which arguably implies large increases in the emissions of ocean sulfur gases via processes that involve ocean productivity. COS and DMS have direct and indirect climate feedbacks that can impact the evolution of deglacial cycles. Better constraints on the atmospheric COS budget, particularly on the specifics of the ocean sources, coupled with a modelling effort are needed to quantitatively partition the necessary emissions increases among different sources and to refine climate

implications.

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

**Supplementary Code.** Sample Stan codes used in the statistical modelling are provided as a supplementary codes S1 and S2 in the supplement document.

**Author contributions.** MA, MRN, ESS, CFL, TH, KJC, DW, DF, EO developed measurement methods and conducted measurements. MA and GLB developed the data analysis methodology. MA wrote the original draft. MA, ESS, GLB, MW, DW, MRN revised and edited.

**Competing interests.** The authors declare no competing interests.

**Acknowledgments**. We thank U.S. Ice Drilling Program and the National Science Foundation Ice Core Facility for drilling and curation of the ice cores. Support was provided by National Science Foundation grants 1443470 and 1917485 and Simons Foundation grants 549931 and 645921.

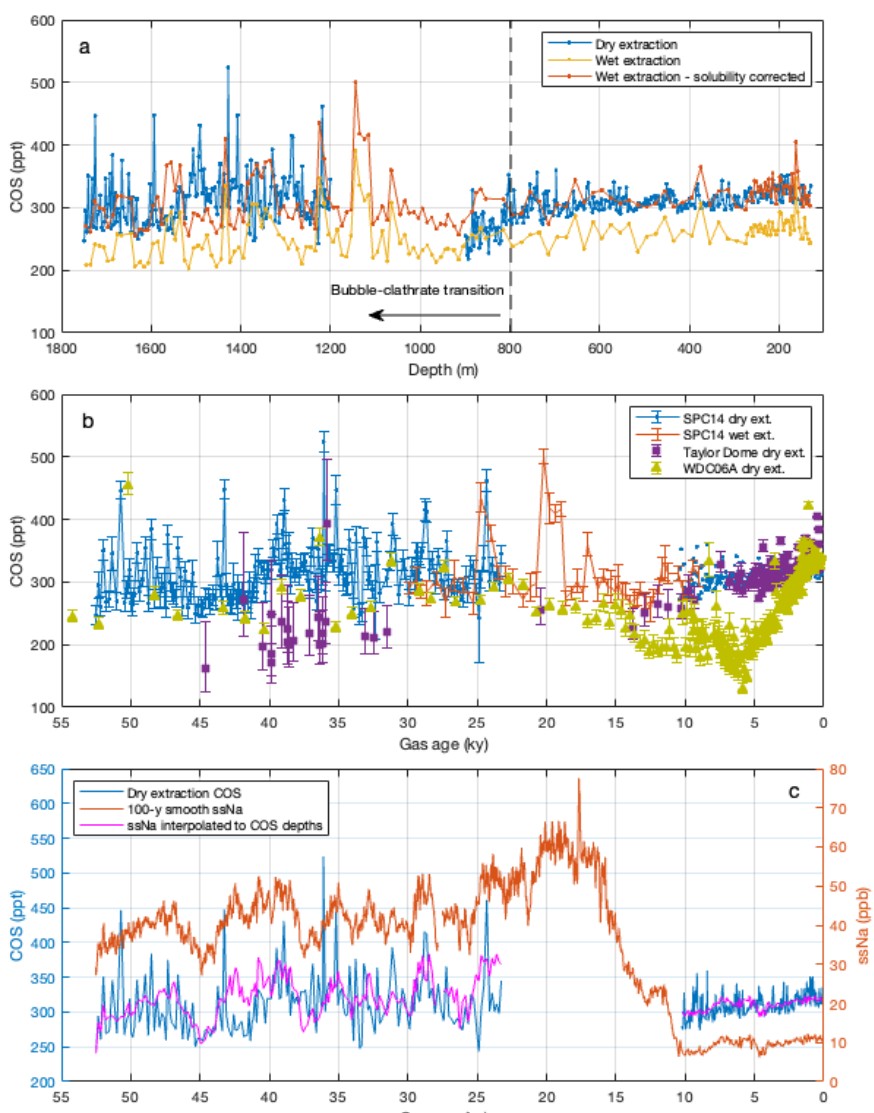

**Figure 1: a)** COS measurements based on dry (blue) and wet (yellow) extraction methods, and solubility-corrected wet extraction data (red) versus depth in the SPC14 ice core. The dry measurements from the BCTZ (dashed line) are excluded from the interpretation. **b)** COS measurements from the SPC14 (blue and red), Taylor Dome (purple), and WAIS Divide (green and yellow) ice cores versus gas age. Only a subset of the wet SPC14 data from 30 – 9 ky is shown because dry measurements are preferred when there is data from both extraction methods. **c)** Dry COS measurements (blue) on the left y-axis and the 100-y smooth ssNa (red) on the right y-axis for the SPC14 ice core. The ssNa interpolated to COS measurement depths (magenta) are shown with an offset of -20 ppb below 1200m and +10 ppb above 800 m.

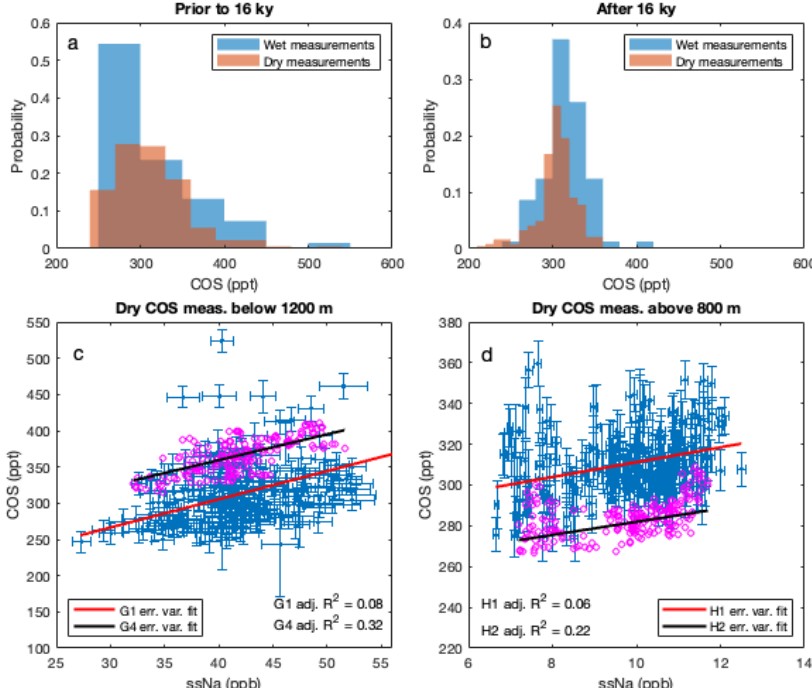

**Figure 2: a)** Distribution of SPC14 COS measurements before 16 ky. The dry measurements shown here are the basis for the regressions shown in (c). **b)** Distribution of SPC14 COS measurements after 16 ky. The dry measurements shown here are the basis for the regressions shown in (d). **c)** The ssNa vs. dry COS measurements for the glacial period (below 1200 m) used in the G1 (blue error bars) and the G4 (magenta circles) scenarios (Table 2). The linear fits to errors-in-variables regressions are calculated using the mean slopes and intercepts for G1 and G4 (also see Figs. A5 and A6). **d)** Same as (c) but for the Holocene (above 800 m) H1 and H2 scenarios (also see Fig. A7).

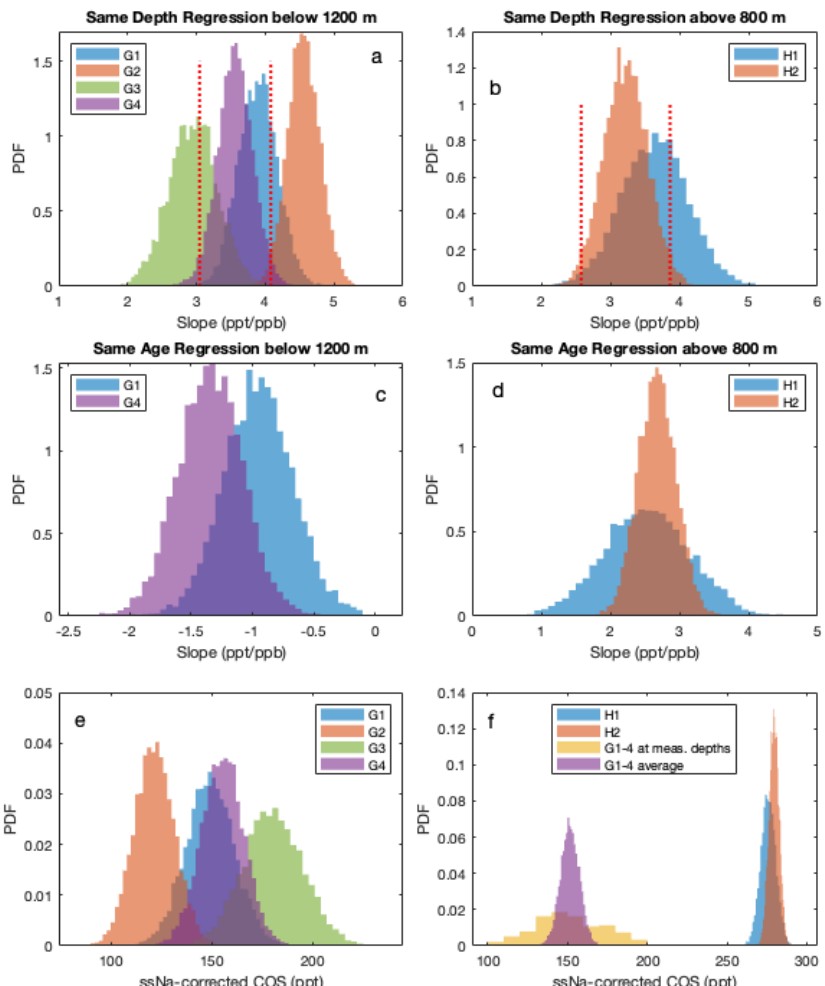

**Figure 3. a)** PDFs of the same-depth errors-in-variables regressions for the glacial period (below 1200 m) dry extraction COS measurements under four different analysis scenarios (section 3.1). **b)** Same as (a) for the two Holocene (above 800 m) scenarios. In (a) and (b), the $\pm 2\sigma$ (95%) confidence intervals (red dashed lines) for the G4 and H2 regressions are shown to demonstrate the high statistical significance (being different from zero) of the PDFs for all scenarios. **c-d)** Same as (a) and (b) but for the same-age errors-in-variables linear regressions. **e)** Distributions of all dry COS measurements from $52 - 23$ ky after correction with ssNa under G1 through G4 scenarios. **f)** Distributions of dry COS measurements after correction with ssNa under H1 (blue) and H2 (red) scenarios. For comparison, distributions of the average across G1 through G4 (purple, n=8000) and the average of all scenarios at discrete depths (yellow, n=177) are also shown. During the LGM, the ssNa-corrected wet COS measurements reach lower levels than what is displayed here for the $52 - 23$ ky period (Fig. 4b).

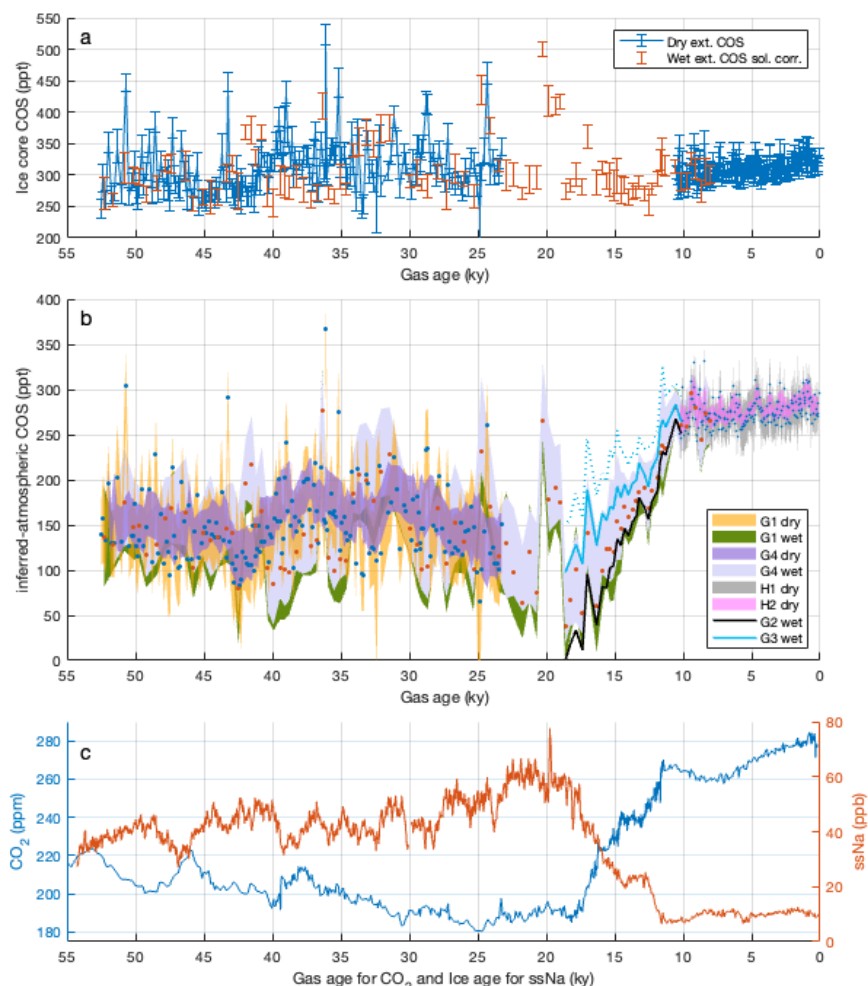


**Figure 4. a)** The SPC14 COS measurements from Fig. 1a with ±1σ measurement errors. **b)** The inferred (ssNa-corrected) atmospheric COS record. The filled markers denote the individual data points corrected with G1 and H1 scenarios for the dry (blue) and G1 for the wet (red) measurements. The colored bands (see legend) denote posterior ±2σ ranges (section 2.4) for all analysis scenarios shown in Fig. 3a-d; the full PDFs for G1 through G4 and H1 and H2 scenarios are shown in Fig. 3e, f.

The wet G1 ±2σ range is mostly masked by the wet G4 ±2σ range. For clarity, only the posterior means (lines) through the deglaciation are shown for the wet G2 and G3. The dashed blue line is the +2σ range for the G3 scenario, representing an upper limit albeit at very low probability. **c)** A composite Antarctic ice core $CO_2$ record (blue – left y-axis) and the 100y-smooth ssNa record from the SPC14 ice core (red – right y-axis) versus gas age and ice age, respectively (Bereiter et al., 2015; Winski et al., 2021).


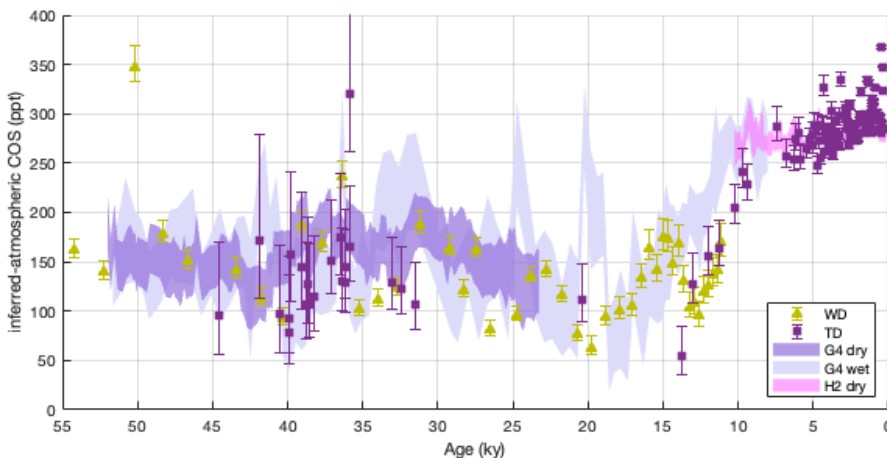

**Figure 5.** The ice core COS records from the TD (purple squares) and WD (green squares) ice cores are corrected for COS production using the G4 ssNa-COS errors-in-variables regression slope. The ssNa-corrected South Pole record for different scenarios (Fig. 4b) is shown in the back ground.


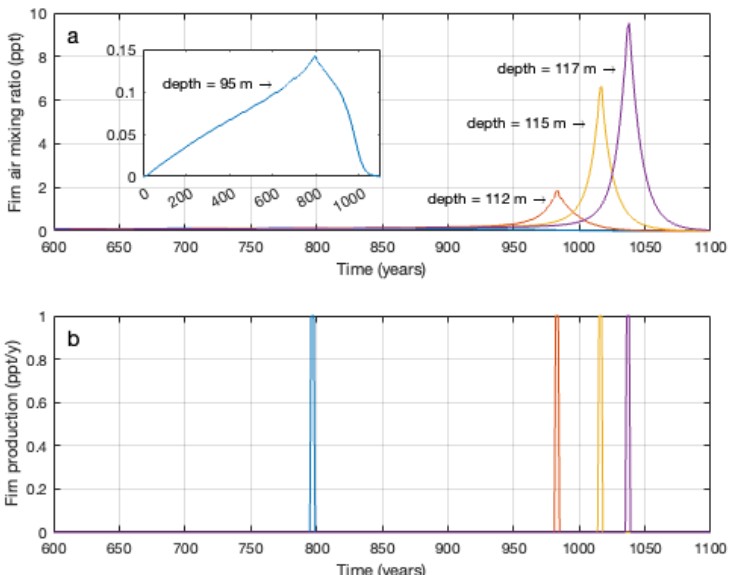

**Figure 6.** Modeled trace gas production in the South Pole firn from an impurity layer. **a)** Firn air trace gas mixing ratio (y-axis) vs time (x-axis) at 95 m (blue), 112 m (red), 115 m (yellow), and 117 m (purple). The 95 m profile is shown in the inset because the excess gas peak at this depth is not visible at the scaling of the parent plot. Note that the gas mixing ratio at any given depth peaks when the production occurs at that depth. **b)** Gas production vs. time at the same depths as (a).

**Table 1.** Atmospheric COS budget.

| | Sources (Gg S y$^{-1}$) | |
|---|---|---|
| | **Bottom-up*** | **Top-down**** |
| **Ocean – direct as COS** | $130 \pm 80$ | 39 |
| **Ocean – indirect as CS$_2$** | $135 \pm 130$ | 156 |
| **Ocean – indirect as DMS** | $53 \pm ?$*** | 81 |
| **Anthropogenic (CS$_2$ + COS)** | $400 \pm 180$ | 180 |
| **Biomass burning** | $116 \pm 52$ | 136 |
| **Anoxic soils and wetlands** | -150 to 290 | |
| **Volcanism** | 25 to 43 | |
| **Unidentified** | | 600**** |
| **Total** | 938 | 1192 |

| | Sinks (Gg S y$^{-1}$) | |
|---|---|---|
| | **Bottom-up** | **Top-down** |
| **Canopy and Soil uptake** | 700 to 1100 | 1093 |
| **OH oxidation** | 80 to 130 | 101 |
| **Stratospheric loss** | $50 \pm 15$ | |
| **Total** | 1055 | 1194 |

* Whelan et al. (2018).
** Berry et al. (2013).
*** Based on mechanistic modelling Jernigan et al., 2022, the uncertainty is not quantified.
**** Presumed photochemical and attributed to low latitude oceans.

**Table 2.** Descriptions of different data analysis scenarios.

| Scenario | COS | ssNa | Alpha | Same-age regression |
|---|---|---|---|---|
| G1 | Measured | 100-y smooth | No | No |
| G2 | 7-point mov. avg. | 1100-y smooth | No | No |
| G3 | 7-point mov. median | 1100-y smooth | Yes | No |
| G4 | 7-point mov. median | 1100-y smooth | Yes | Yes |
| H1 | Measured | 100-y smooth | No | No |
| H2 | 7-point mov. avg | 300-y smooth | Yes | Yes |

## Appendix A: Supplementary figures

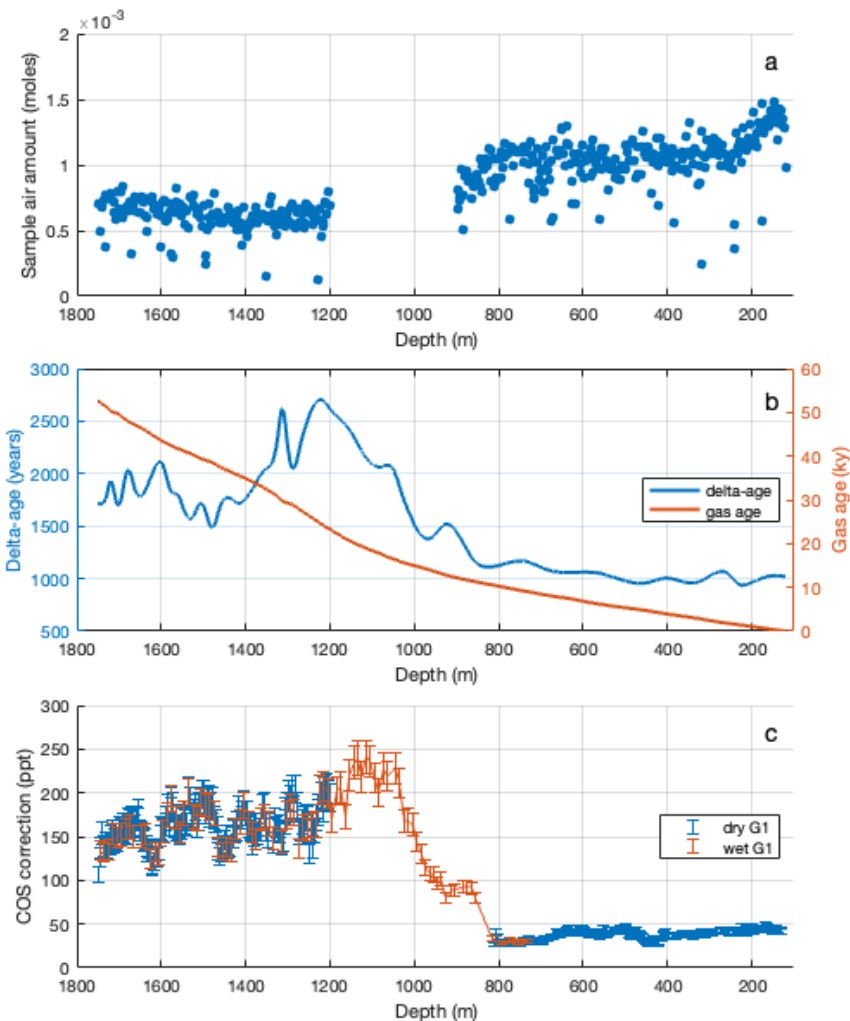


**Figure A1. a)** Amount of air extracted from the SPC14 ice core samples used for COS analysis with dry extraction. The low outliers are due to smaller ice samples that were commonly half as long. Average amount of air extracted from samples below 1200 m is lower primarily because of a drop in extraction efficiency of dry extraction for full clathrate ice because of the smaller size of clathrates with respect to the bubbles. The presumed bubble-clathrate transition zone from 900 m to 1200 m

was not analyzed with dry extraction because of expected experimental artifacts. The actual start depth of BCTZ is around 800 m as evidenced by the downcore decrease in sample air amount. **b)** The delta-age (left y-axis) and gas age (right y-axis) for the SPC14 ice core (Epifanio et al., 2020). **c)** The correction applied to the COS measurements under the G1 scenario.

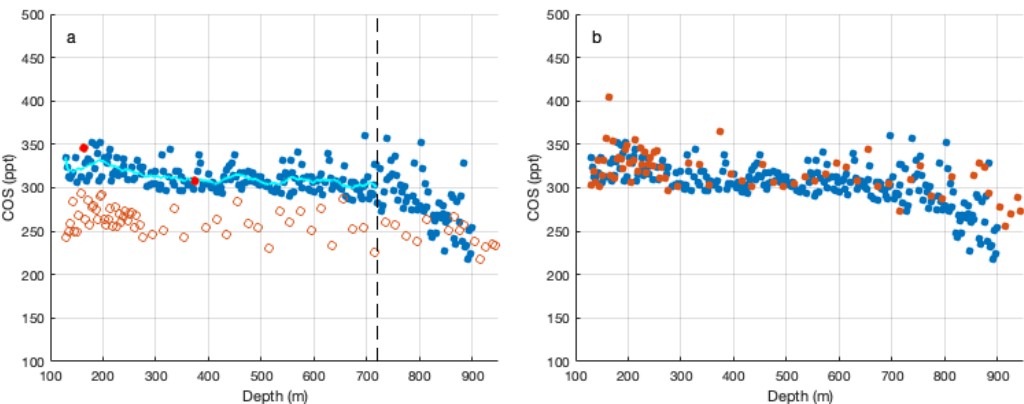

**Figure A2.** The dry (blue) and wet (red) extraction based COS measurements from the top 950 m of the SPC14 ice core. **a)** The wet extraction measurements from above 720 m (shallower than vertical dashed line) are smoothed before being used to determine the solubility correction parameters (Fig. A3). The smooth dry extraction record (cyan line) is linearly interpolated to wet extraction measurement depths to estimate $COS_{dry}$ in Eq. (3). Solid red circles denote two outlier wet extraction measurements, which were not included in determination of $C_3$ and $C_4$ in Eq. (3). **b)** Solubility corrected wet extraction

measurements (red) versus the dry extraction measurements from (a).

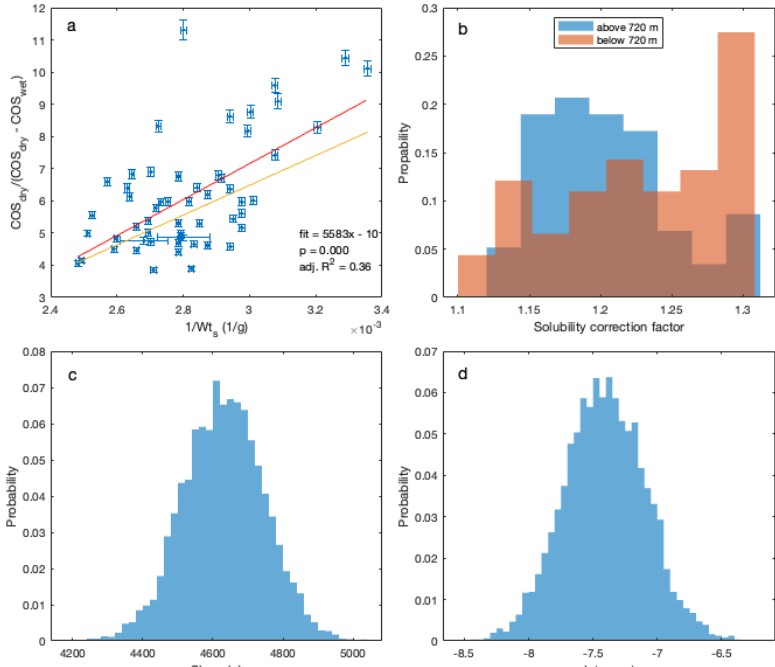

**Figure A3.** Solubility correction parameters and the probability distributions functions (PDFs) of solubility correction factors.
**a)** Least squares (red line) and errors-in-variables (yellow line) regressions of equation-3. The slope (5583 [g]) and the intercept
(-10 [unitless]) for the least squares regressions are shown for comparison with the errors-in-variables regression parameters
in (c) and (d). **b)** The histograms of solubility correction applied to the wet extraction COS measurements. **c)** Probability
distribution of errors-in-variables regression. **d)** Probability distribution of the intercept of errors-in-variables regression.

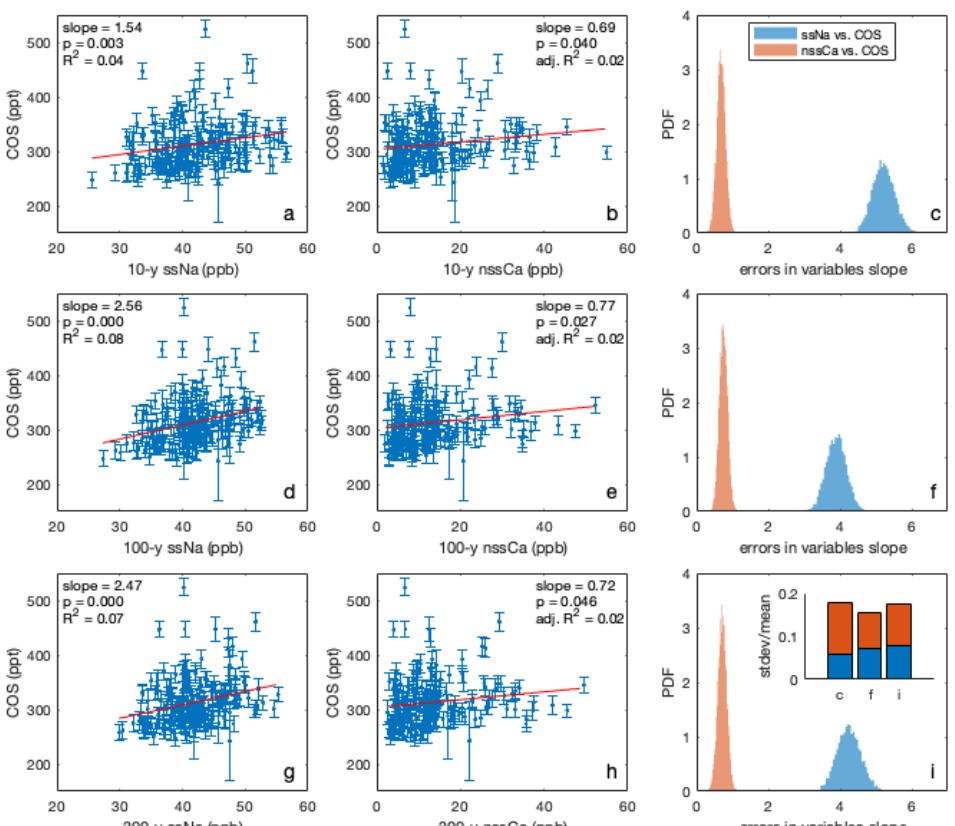

**Figure A4.** Same-depth regression analyses between ssNa and nssCa versus dry extraction COS below 1200 m. **a)** 10-y ssNa versus COS with least-squares regression results. **b)** 10-y nssCa versus COS with least-squares regression results. **c)** Errors-in-variables regression results for a and b. **d-e)** Same as (a) and (b) but for 100-y ssNa and nssCa. **f)** Errors-in-variables regression results for (d) and (e). **g-h)** Same as (a) and (b) but for 300-y ssNa and nssCa. **i)** Errors-in-variables regression results for g and h. The inset displays stdev/mean (a significance metric) for the PDFs shown in c, f, and i.


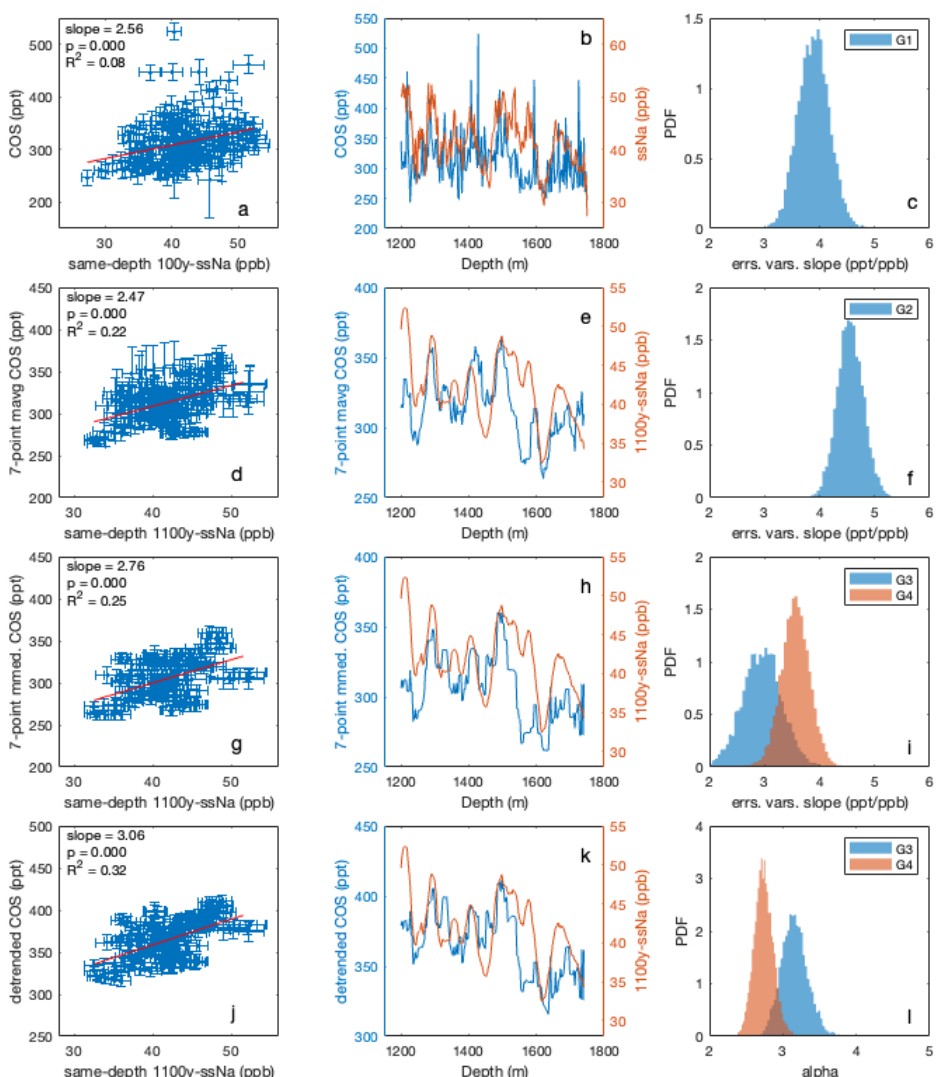

**Figure A5.** Same-depth correlations between ssNa and COS for different glacial period (>1200 m) analysis scenarios. **a)** G1: 100-y ssNa versus COS with least-squares regression results (same as Fig. 2c). **b)** Data from (a) vs. depth. **c)** Errors-in-variables PDF for (a) (same as in Fig. A4f). **d)** G2: 1100-y ssNa versus 7-point moving average COS with least-squares regression results. **e)** Data from (d) vs. depth. **f)** Errors-in-variables PDF for (d). **g)** G3: 1100-y ssNa versus 7-point moving median COS with least-squares regression results. **h)** Data from (g) vs. depth. **i)** Errors-in-variables PDFs for (g) and (j). **j)** G4: 1100-y ssNa versus 7-point moving median COS detrended for same-age correlation with ssNa, shown with least-squares regression results. **k)** Data from (j) vs. depth. **l)** Errors-in-variables PDFs for the G3 and G4 alpha parameters.

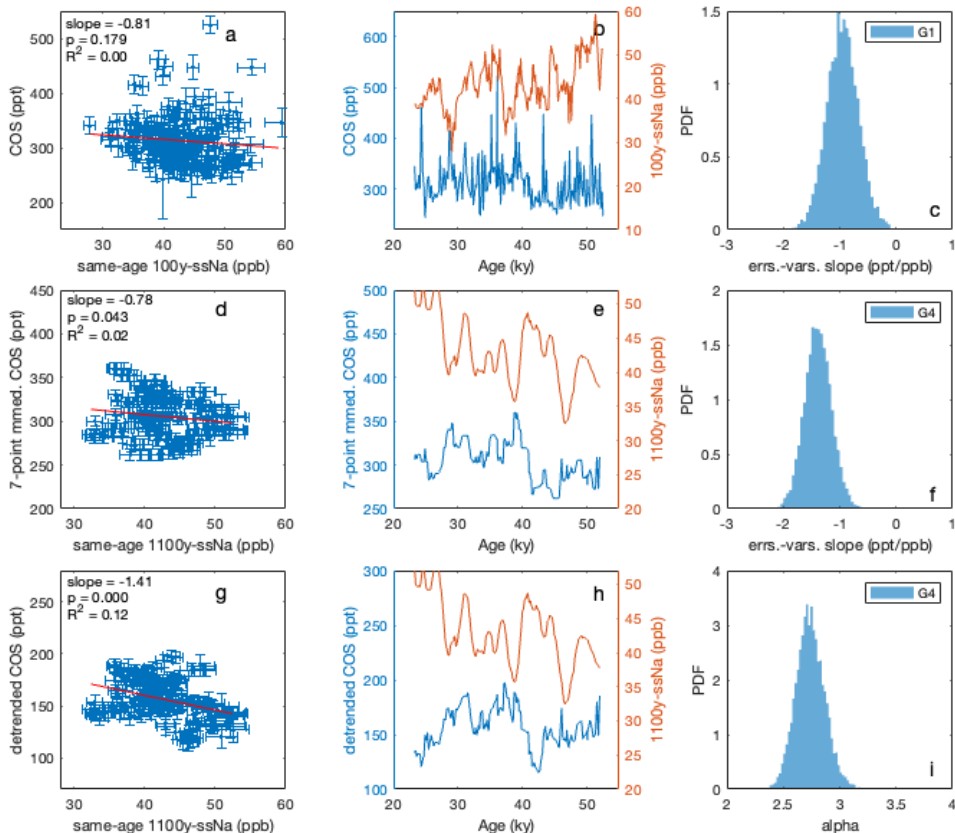


**Figure A6.** Same-age correlations between ssNa and COS for different glacial period (>1200 m) analysis scenarios. **a)** Least squares regression of 100-y ssNa versus COS. **b)** Data from (a) vs. age. c)  G1: Errors-in-variables PDF for data shown in (a). **d)** Least squares regression of 1100-y ssNa versus 7-point moving median COS. **e)** Data from (d) vs. age. **f)** G4: Errors-in-variables PDF for data shown in (g). **g)** Least squares regression of 1100-y ssNa versus 7-point moving median COS detrended for same-depth correlation. **h)** Data from (g) vs. age. **i)** Errors-in-variables PDF for the G4 alpha parameter (same as in Fig. A5l).


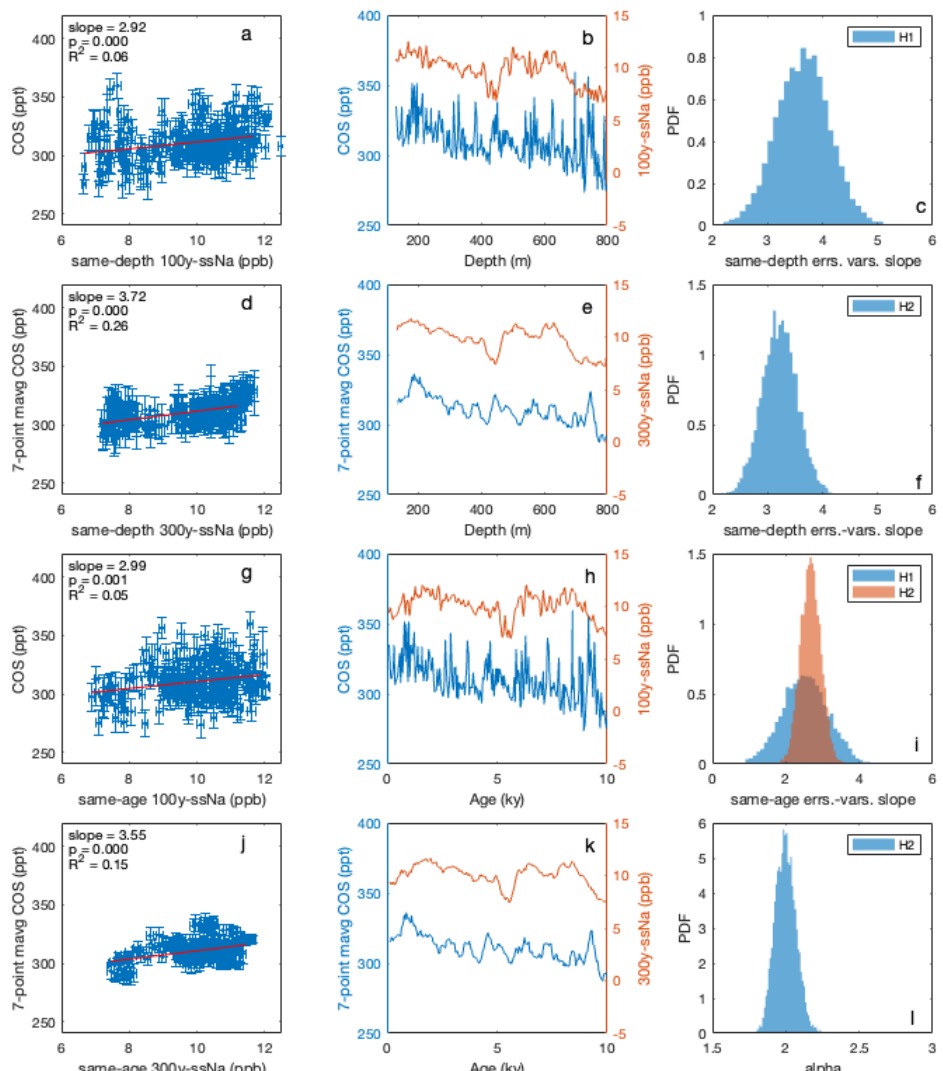

**Figure A7.** Same-depth and same-age correlations between ssNa and COS for different Holocene (< 800 m) analysis scenarios. **a)** H1: Same-depth 100-y ssNa versus COS with least-squares regression results. **b)** The data from (a) shown versus depth. **c)** Errors-in-variables PDF for (a). **d)** H2: Same-depth 300-y ssNa versus 7-point moving average COS with least-squares regression results. **e)** The data from (d) shown versus depth. **f)** Errors-in-variables PDF for (d). **g)** H1: Same-age 100-y ssNa versus COS with least-squares regression results. **h)** The data from (g) shown versus age. **i)** Errors-in-variables PDFs for (g) and (j). **j)** H2: Same-age 300-y ssNa versus COS with least-squares regression results. **k)** The data from (j) shown versus age. **l)** Errors-in-variables PDF for the H2 alpha parameter.

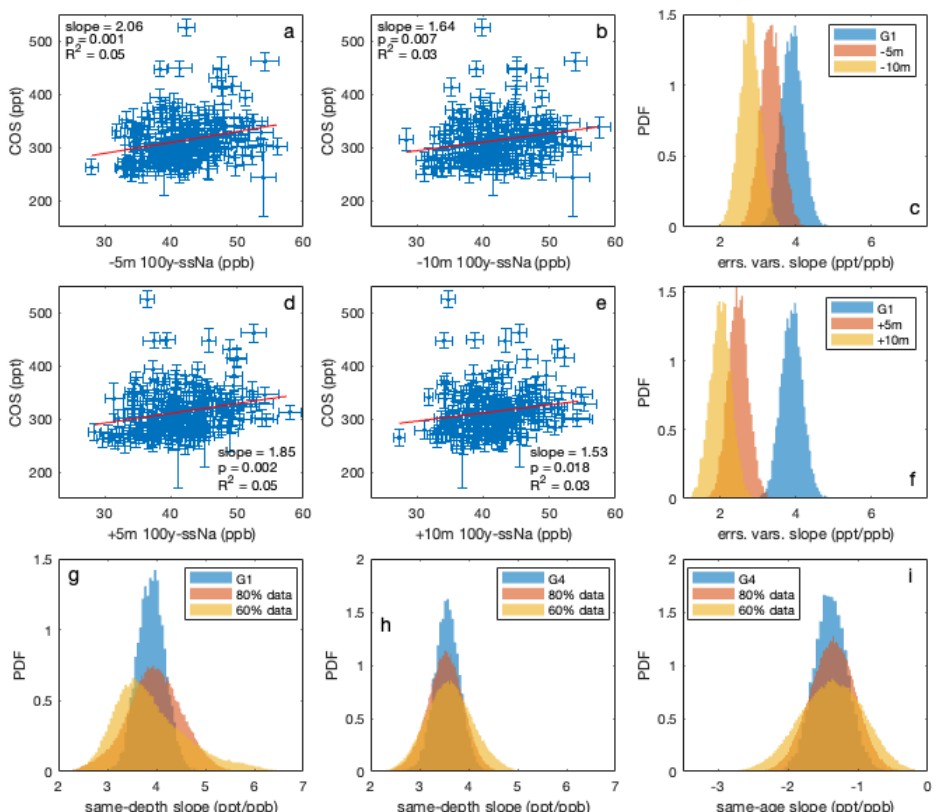

**Figure A8. Sensitivity tests for G1 and G4 analysis scenarios. a)** 100-y ssNa vs COS correlation with least-squares regression results. ssNa is offset from COS measurement depths by -5 m. **b)** Same as (a), but ssNa is offset from COS measurement depths by -10 m. **c)** The errors-in-variables PDFs for (a) and (b) along with the G1 same-depth PDF from Fig. A5c. **d, e, f)** Same as (a), (b), (c), but the offsets are +5 m and +10m. **g)** Repeat of the G1 same-depth analysis by random sampling of the original data set at 80% and 60% in 20 simulations each. **h)** Repeat of the G4 same-depth analysis by random sampling of the original data set at 80% and 60% in 20 simulations each. **i)** Repeat of G4 same-age analysis by random sampling of the original data set at 80% and 60% in 20 simulations each.


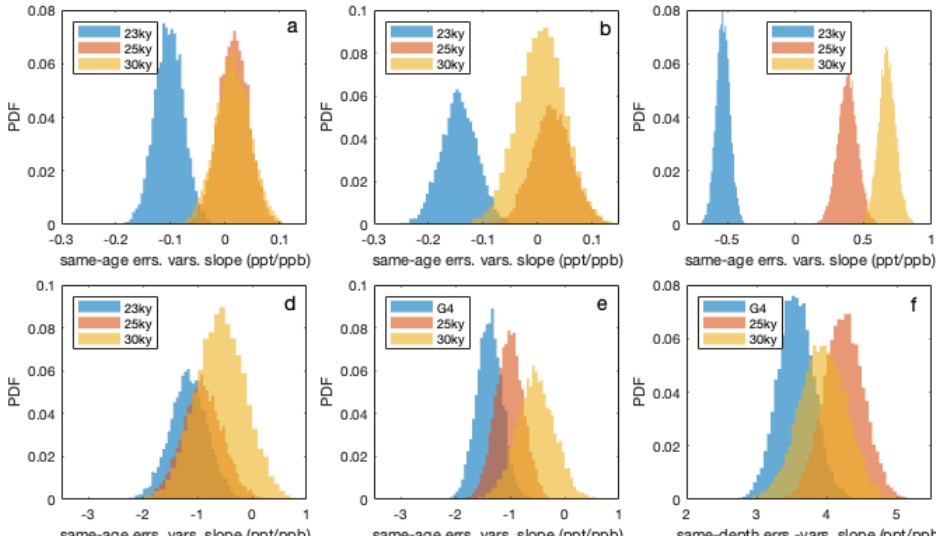

**Figure A9. a)** The errors-in-variables PDFs for the regression of the inverted 100-y ssNa interpolated to the SPC14 gas chronology at annual resolution versus the 100-y ssNa record on ice chronology at annual resolution during three periods (>23 ky, >25 ky, and >30 ky). **b)** Same as (a) except using 1100-y ssNa. **c)** Same as (b) but at the ages of the COS measurements. **d)** The errors-in-variables PDFs for the regression of the mean of the G4 corrected COS from Fig. 4b with the 1100-y ssNa. **e)** The same-age results from a repeat of the G4 regression (Fig. 3c) for different temporal cut-offs. **f)** The same-depth results from a repeat of the G4 regression (Fig. 3a) for different temporal cut-offs.

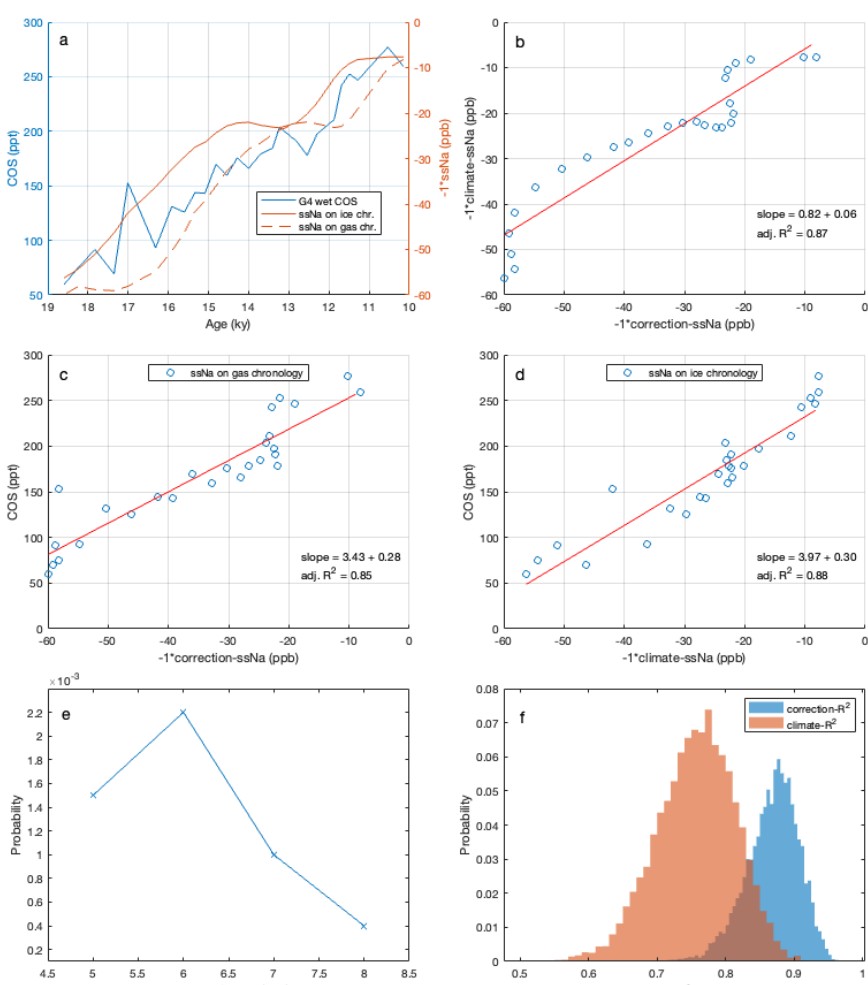

**Figure A10. a)** The corrected COS (G4) during the deglaciation (left y-axis), and the correction-ssNa and climate-ssNa (right y-axis). **b)** The regression of the correction-ssNa (x-axis) versus the climate-ssNa (y-axis). **c)** The regression of the correction-ssNa versus COS. **d)** The regression of the climate-ssNa versus COS. **e)** The conditional probability of the climate-$R^2$ being higher than the correction-$R^2$, given the climate-$R^2$ is at least 0.88, at different levels of random noise. **f)** The PDFs of the correction-$R^2$ and the climate-$R^2$ at 6 ppb noise.


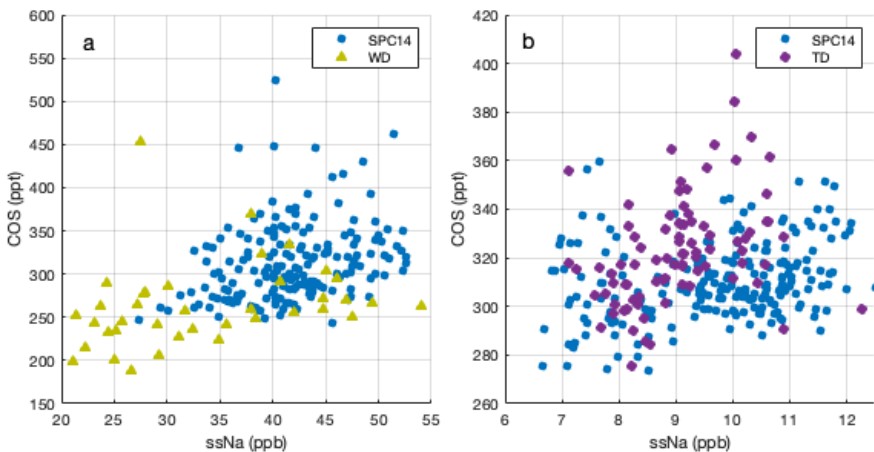

**Figure A11.** 100-y ssNa vs. measured COS levels. ssNa for the WD and TD ice cores is calculated using $Na^+$ and $Ca^{2+}$ measurements from the respective cores (McConnell et al., 2017; Steig et al., 2000). **a)** The data used in the G1 scenario (Fig. 2c) for SPC14 is compared with all data from WD older than 11 ky. **b)** The data used in the H1 scenario (Fig. 2d) for SPC14 is compared with TD data younger than 8 ky.

## Appendix B: Sensitivity tests for G1 and G4 scenarios

We conducted two sensitivity tests for G1 and G4 scenarios. The first test is designed to test whether ssNa interpolated to the same depth as the COS samples is indeed the best predictor of the COS production. We repeat the G1 analysis with 100-y smooth ssNa interpolated to 5 m and 10 m above and below the actual COS measurement depths (Fig. A8a-f). The significance of the ssNa-COS relationship deteriorates (PDFs move closer to zero) with offsets in either direction, confirming that ssNa interpolated to the same depths as the COS measurements is the best predictor of the COS production although this evidence alone is not sufficient to attribute the inferred production to a specific depth range in the firn.

In the second test, we explore the sensitivity of the regression results to random elimination of a subset of the measurements from the regressions. We repeat the G1 and G4 analyses with 20 subsets of the original COS data set (total of 40 simulations). Data in each subset is randomly picked without replacement to sample 80% and 60% of the full data set. In G1 (Fig. A8g), the mean slope value of the same-depth regression is not sensitive to the reduction in the amount of data included in the analysis but the shape of the distribution and the significance are. In G4 (Fig. A8h, i), the simultaneous same-depth and same-age regression results are robust with respect to the reduction in the data amount, with some reduction in significance at 20% data reduction and more at 40%, but with minimal impact on the mean value and the shape of the distribution. These results demonstrate G4 results are more robust with respect to data density than the G1.

## Appendix C: The validity of the glacial period same-age anticorrelation between ssNa and COS as a climate signal

The same-age correlation analyses display an anti-correlation between ssNa and COS during the glacial period (52 – 23 ky) even when such analyses are conducted independently between measured COS and ssNa without a simultaneous same-depth correction. Simultaneous same-depth regression strengthens this relationship (Fig. 3c). Here, we explore whether this strengthening could be resulting from the fact that the applied same-depth correction itself is anti-correlated with the ssNa record. These tests are akin to testing the autocorrelation of the ssNa record at a given lag, but instead of a fixed lag, the delta-age for the ice core is used.

We set up three control scenarios. In the first scenario, we interpolate an inverted (multiplied by -1 to simulate the ssNa correction to the COS record) 100-y smooth ssNa record to the SPC14 gas chronology at annual resolution, then regress it against the original 100-y ssNa record on ice chronology for the same depths over periods older than 23 ky, 25 ky, and 30 ky (Fig. A9a). The second scenario is identical to the first except we used the 1100-y smooth ssNa record (Fig. A9b). The third scenario is a repeat of the second at lower and irregular resolution, using only the 1100-y ssNa data interpolated to the COS measurement depths (Fig. A9c). All three control scenarios reveal significant but small negative slopes for the 23 ky cut-off, but truncating the analyses with the 25 ky or 30 ky cut-offs results in either no significant correlation or a positive correlation.

The three control scenarios can be compared with the same-age regressions between ssNa and COS corrected for excess COS using the same-depth relationship under G4. When the corrected COS from G4 is regressed with the same-age 1100-y ssNa, we find that it is more robust against the temporal cut-offs (Fig. A9d). This is slightly different from what is done in G4

because we do not conduct a simultaneous same-depth regression. Repeats of the G4 regression itself using different temporal cut-offs reaffirm that the same-age anticorrelation between ssNa and COS during 52 – 23 ky results primarily from variance in the measured COS mixing ratios and not from the applied same-depth correction (Fig. A9e). Further, the same-depth regression results do not vary significantly or show any sign of deterioration at different temporal cut-offs (Fig. A9f).

**Appendix D: The COS rise concurrent with the deglaciation**

Here, we investigate whether there is any explicit statistical evidence that the increase in the inferred atmospheric COS record concurrent with the deglaciation during 19-10 ky is climate driven. The same-depth ssNa correction alone would result in a temporal COS trend that looks like the inverted ssNa record on the gas age scale (correction-ssNa) whereas the true climate signal in the ssNa record is evident on the ice chronology (climate-ssNa). The corrected COS record from G4 is regressed against the 1100-y smooth (G4) correction-ssNa and against the climate-ssNa (Fig. A10). The $R^2$ statistic for the regression versus the correction-ssNa is 0.85 (Fig. A10c), indicating the magnitude of the deglacial COS rise is set primarily by the ssNa correction. The $R^2$ of the regression versus the climate-ssNa should be comparatively lower if the COS measurements merely introduce random noise into the corrected COS record, but the climate-$R^2$ is 0.88 and higher than the correction-$R^2$ (Fig. A10c vs. d).

It is possible to estimate the probability that we would observe these climate-$R^2$ and correction-$R^2$ values coincidentally. To do so, we introduce random noise to the correction-ssNa, then regress against the actual correction-ssNa (itself without the noise), and separately against the climate-ssNa (also without noise) in 10,000 simulations, calculating the correction-$R^2$ and climate-$R^2$ for each instance. We find that the conditional probability of climate-$R^2$ being higher than the correction-$R^2$, given that climate-$R^2$ is equal to or higher than 0.88, peaks at about p=0.002 with 6 ppb (±1stdev) of added noise (Fig. A11e). The probabilities are lower at lower noise because the correction-$R^2$ approaches 1, and also lower at higher noise because the climate-$R^2$ decreases rapidly. It is highly probable that the wet COS measurements from the deglaciation include a climate signal, which is retained during the same-depth ssNa correction, resulting in an inferred atmospheric COS record that is more similar to the deglacial climate signal than the inverse of the applied ssNa correction.

**Appendix E: Temperature and pH effects on gas solubility and COS hydrolysis**

Solubility of gases in the ocean are temperature dependent. Sea surface temperature rises during the deglaciation and this effect alone should result in an increase in gas flux out of the ocean due to decreased solubility. A +1°C change in temperature results in less than 5% decrease in the solubilities of COS, $CS_2$, and DMS over a range of 0 to 30 °C (DeBruyn et al., 1995; Elliot et al., 1989). Present-day ocean $CS_2$ flux correlates with temperature and can increase by about 10% per °C (Xie and Moore, 1999); this effect is too high to be solely due to the temperature effect on $CS_2$ solubility. Direct COS emissions are buffered against temperature increases because as solubility decreases, the hydrolysis loss rate increases at a faster rate, with +1°C change in temperature resulting in about a 10% rise in the COS hydrolysis rate constant. The hydrolysis of COS is

also sensitive to pH and slows down by about 5% per 0.1 unit decrease at pH 8 (Elliot et al., 1989). For the direct COS flux, the net impact of a +1°C temperature increase coupled with a 0.1 unit decrease in pH is close to zero. It is not possible to drive large changes in ocean-atmosphere gas fluxes via temperature effects on gas solubility, or via the coupled effects of temperature and pH on COS hydrolysis.

**Appendix F: Calculation of the COS radiative impact**

Brühl et al. (2012) calculated a radiative impact of -0.007 $Wm^{-2}$ for about a 30% (roughly 150 ppt) increase in atmospheric COS during the 20th century due to anthropogenic emissions. They also estimate that the radiative impact of the same amount of COS as a greenhouse gas in troposphere is 0.003 $Wm^{-2}$, yielding a net impact of -0.004 $Wm^{-2}$.

The radiative impact of increased atmospheric COS depends on the emission geometry. This has been estimated in chemistry-climate model experiments with two distinct emission geometries (Quaglia et al., 2022). In the first model experiment, COS was injected into the atmosphere using the geographic distribution of present-day anthropogenic emissions at the surface, thus the emissions occur primarily over land in the northern hemisphere where the terrestrial uptake is also strong. In the second model experiment, COS was injected at the tropical tropopause, which enables quick dispersion into the stratosphere. The location of the emissions impacts the relative increase in the tropospheric versus the stratospheric mixing ratios of COS. In the troposphere, COS acts primarily as a greenhouse gas and has a positive radiative impact. In the stratosphere, COS acts primarily as a source of sulfate aerosol with a negative radiative impact. The model also incorporates indirect radiative effects via ozone, methane, and stratospheric water vapor.

The net (tropospheric + stratospheric) radiative impact of a 40 TgS $y^{-1}$ emission increase in the first experiment is -1.3 $Wm^{-2}$. The net radiative impact of a 6 TgS $y^{-1}$ emission increase in the second experiment is -1.5 $Wm^{-2}$. In the first and second modeling experiments, the tropospheric mixing ratio of COS stabilizes at 35 ppb and 5 ppb, respectively. Assuming linear scaling, these sensitivities imply -0.006 $Wm^{-2}$ (first experiment) and -0.05 $Wm^{-2}$ (second experiment) net radiative impacts for a 150 ppt increase in tropospheric COS. The deglacial increase in COS emissions is largely oceanic in origin, hence primarily in the boundary layer and in the southern hemisphere. The radiative impact estimates from the two model experiments should be considered unrealistic lower and upper bounds. The best estimate may be closer to the lower bound estimate from the first model experiment.