# Peer review of "Carbonyl sulfide measurements from a South Pole ice core and implications for atmospheric variability since the last glacial period"

_Climate of the Past, 2023_

## Referee Comment (RC1)

Derivation of Eq. (6):

$$\begin{cases} \dfrac{ssCa^{2+}}{ssNa^+} = R_m \\ \dfrac{nssCa^{2+}}{nssNa^+} = R_t \end{cases}$$

$\Rightarrow$

$$\begin{aligned} Ca^{2+} &= ssCa^{2+} + nssCa^{2+} \\ &= R_m ssNa^+ + R_t nssNa^+ \\ &= R_m ssNa^+ + R_t \left( Na^+ - ssNa^+ \right) \end{aligned}$$

$\Rightarrow$

$$(R_t - R_m) ssNa^+ = R_t Na^+ - Ca^{2+}$$

$$\Rightarrow$$

$$\text{ssNa}^+ = \frac{R_\text{t}\text{Na}^+ - \text{Ca}^{2+}}{R_\text{t} - R_\text{m}} = \frac{\text{Na}^+ - \dfrac{\text{Ca}^{2+}}{R_\text{t}}}{1 - \dfrac{R_\text{m}}{R_\text{t}}}$$

$$\Rightarrow$$

$$\begin{aligned}
\text{nssCa}^{2+} &= R_\text{t}\left(\text{Na}^+ - \text{ssNa}^+\right) \\
&= R_\text{t}\text{Na}^+ - \frac{R_\text{t}\text{Na}^+ - \text{Ca}^{2+}}{1 - \dfrac{R_\text{m}}{R_\text{t}}} \\
&= \frac{\text{Ca}^{2+} - R_\text{m}\text{Na}^+}{1 - \dfrac{R_\text{m}}{R_\text{t}}}
\end{aligned}$$

---

## Author Comment (AC1)

The reviewer comments are in black font and our responses are in blue.

**Reply to Wu Sun (reviewer #1)**

In this work, Aydin et al. painstakingly recovered a 52,500-year record of atmospheric carbonyl sulfide from a South Pole ice core, with great care and attention devoted to correcting for the post-depositional COS production from sea salt aerosols and other artifacts during extraction. The resulting data set is a valuable contribution to the atmospheric history of carbonyl sulfide and will enable the climate and ecosystem modeling community to better understand biospheric changes since the last ice age.

Despite its scientific significance, the organization of the manuscript may hinder its key findings from being grasped by a broad audience. For this reason, I suggest clarifying the methods, especially the rationale behind every correction and sensitivity test, and streamlining the presentation of the main messages to strengthen the work. Below I list a few high-level issues followed by specific line-by-line comments.

My issues with the methods are:

- For the calculation of ssNa$^+$ (Sect. 2.3), please check the denominator in the right hand side of Eq. (6). My derivation says it should be "$1 - R_m/R_t$" in the denominator (see attached slides). In other words, there is no reason for Eqs. (6) and (7) to have different denominators.

The reviewer is correct. Both denominators should display a subtraction of Rm/Rt from 1. This is now corrected. This change results in a linear 4% increase in the calculated ssNa but no change in the data analysis.

- For the regression between COS and ssNa$^+$ (Sect. 2.4), it was not clear why errors in the response variable need to be scaled with a multiplier (α in Eq. 8). This multiplier did not appear in Eq. (5).

We included alpha as a multiplicative scaler in eq. 8 to demonstrate the impact of a possible bias in determination of the measurement errors (e.g. consistent underestimation of the measurement errors) on the regression results. The alpha parameter is 1 for three of the scenarios (G1, G2, H1) and its distribution is determined by the Bayesian algorithm in the other scenarios, allowing us to demonstrate the difference between using calculated measurement errors versus what the impact on the results is if the measurement errors are biased low. We included new text in section 2.4 to clarify (P6, L195-196): "Inclusion of the $\alpha$ parameter allows quantification of the sensitivity of the correction slope to possible bias of COS measurement errors."

- It was also unclear how measurement errors in COS and ssNa$^+$ (Sect. 2.5) were incorporated into the Bayesian errors-in-variables regression between COS and ssNa$^+$ (Sect. 2.4).

The COS errors are directly used as measurement errors (i.e. errors in y, denoted yerr in equations 8, 9, and 10). The ssNa errors are used as xerr in eq. 4. We added text to section 2.4 to clarify (P7 L201-202): "The error estimates for the COS and ssNa measurements are directly incorporated in the errors-in-variables regressions as they substitute for y$_{err}$ and x$_{err}$ in equations

4 and 8, allowing us to propagate the uncertainty in the measurements to the slope of the relationship between ssNa-COS."

- Different scenarios for correcting for ssNa$^+$-produced excess COS (Table 1) should be clearly outlined in the methods section, with the rationale explained. It was not until I read the results did I get a hint of why these scenarios were needed and why they were set up like these. The choice of the averaging window size appears ad hoc.

The manuscript is revised to introduce the different scenarios under methods in section 2.4 with a sufficient amount of detail about their purpose (P6 180-189). Some details of the scenarios are still presented under results in section 3.1 because decoupling some of the descriptions from the results of the scenarios makes the text incomprehensible.

The reviewer is correct that the exact values of the averaging windows are ad hoc. The 100-y averaging window is chosen to be higher than the span of the ice age of a typical sample (about 10 y during the glacial period) and yield a higher statistical significance. The 1100-y window is chosen to match the averaging window of the smoothing applied to the COS record to diminish the influence of the spikes on the regressions. The results are robust with respect to the averaging windows despite the large range explored in data averaging.

- Too many scenarios overlaid on the same plot (e.g., Figs. 4b and 5) make it hard to discern the key information. Consider presenting only scenarios that are most robust and relevant to the interpretations.

The results from all scenarios are displayed together in Fig. 4b because one of points we are trying to get across is the fact that the interpretations rely only on the features that are common to all scenarios: low during the glacial period, lowest during the LGM, and high during the Holocene. In the original version of the manuscript, Fig. 5 included two less scenarios than Fig. 4b. In the revised version, we eliminated three other scenarios (two from glacial and one from the Holocene) from Fig. 5, reducing the number of scenarios displayed in this figure to three. Fig. 5 is now a fairly plain figure.

The results section needs to provide answers to questions raised in the introduction. Currently it is a mix of methods, results, technical arguments, and discussion, making it challenging to navigate. The main issues are:

- Many paragraphs describe the nitty-gritty details of various corrections and make reference to figures in the appendices. Important as they are to ensuring data quality, these are probably not the high-level findings you want the readers to walk away with. Consider moving them to methods or the appendices.
- Descriptions of the correction for ssNa$^+$-produced excess COS are scattered throughout the results section, making it difficult to follow. Consider consolidating the main points about this under a single subsection and offload nonessential details to supplementary materials.

Based on the reviewer comments, what used to be sections 3.2, 3.3, and 3.4 are now moved to the appendices. These sections included many of the additional data analyses including sensitivity tests for G1 and G4 scenarios, testing the validity of the glacial same-age correlation as a climate signal, and statistical tests regarding the COS rise concurrent with the deglaciation. Atmospheric COS variability inferred from different scenarios are still presented under results in

section 3.1 with a modified title "Results of different analysis scenarios: Inferred atmospheric COS variability after the ssNa correction." Section 3.1 contains the details of exactly what was done under each analysis scenario.

- It was not clear how the "climate-driven" correlation between ssNa$^+$ and COS was disentangled from the production artifact. Shouldn't the ssNa$^+$COS production correction be applied first before one can make any robust inference of a climate-driven relationship?

By definition, the climate relationship is observed over time (same-age) and the production relationship is observed over same-depth. If the delta-age is large enough, which is the case for the South Pole ice core, separate same-depth and same-age regressions can reveal both the production and climate driven signal as we show in the manuscript with the independent same-depth and same-age analyses for G1 and H1 scenarios. In a typical data analysis approach, untangling the two relationships from each other would commonly be accomplished by correcting for the production first, followed by quantifying and accounting for the same-age relationship, then cycling back to the same-depth correction, and so forth until the successive corrections do not statistically differ from each other. The simultaneous correction algorithm we deploy here in G4 and H2 scenarios allows carrying out this operation in one step within the Bayesian framework. Note that the software used in this analysis is publicly available and the code is made available as supplemental information. The unique aspect of the code used in this analysis is that it does not make use of an iterative process to accomplish this task.

- West Antarctica and Taylor Dome ice cores (L233–243) need to be described in the methods.

We realize that the names of the ice cores (i.e. WAIS Divide and Taylor Dome) were not explicitly mentioned in the introduction paragraph, likely obscuring the fact that the measurements from these ice cores were presented in prior publications. In the revised version, we modified the introduction paragraph to refer to these ice cores by their names (L50-51) where we describe the findings from these two ice cores, citing the relevant references (L48-55). The references are also provided in the results section where we refer to the results from these ice cores (P8 L263-266) and then again in section 3.2 where the records from different cores are compared with each other (P12 L431). The measurements from these ice cores are published in archival journals including all relevant methods.

With respect to the interpretations of the data, there are a few issues:

- The measurements presented here only show ssNa$^+$ but say nothing about what the anions in sea salts are. Thus, it is unclear whether sea salt aerosols contain *organic* sulfur compounds and the analogy between COS dark production in the ocean and that in ice (L462–L476) does not seem to be empirically supported. Experimental evidence suggests that COS is produced from organic sulfur compounds (Modiri Gharehveran and Shah, 2018, https://doi.org/10.1021/acs.est.8b01618) such as cysteine, methionine, and chromophoric dissolved organic matter (CDOM). In any case, a reduced sulfur precursor is needed to produce COS in the absence of sulfate-reducing microbes.

The most abundant anion in the ice core is Cl(-), which is strongly correlated with Na(+) since Cl(-) is sourced primarily from NaCl. Sulfate is another important anion but is sourced primarily

from inorganic sulfur (SO2) with an additional volcanic component and experiences some post-depositional processing. We conducted regression analyses between COS and all major ions measured in the South Pole ice core, including Cl(-) and SO4(2-). The Cl(-) regression results display linear relationships with COS much like Na(+) for the reason stated above. There is no relationship between COS and SO4(2-). We do not include these other regression analyses in the manuscript because they do not provide additional insight. There are no measurements of organic sulfur from the South Pole ice core that can be used in our analyses, and as far as we know, there are no commonly accepted proxies for organic sulfur content in ice cores.

We are aware that an organic sulfur source is needed for abiotic COS production, e.g. (L619-621): "The commonly postulated abiotic process involves reactions between carbonyl groups and thiyl radicals derived from organic sulfur (Flöck et al., 1997; Modiri Gharehveran and Shah, 2018; Lennartz et al., 2017; 2019; Pos et al., 1998; Zepp and Andreae, 1994)." We do not make any claims of full empirical support for the proposed mechanism, e.g (L605-606): "In the absence of auxiliary data, we can only speculate on viable mechanisms for the COS production in the firn." To clarify, this sentence has been revised to read "In the absence of direct empirical evidence, we can only speculate on viable mechanisms for the COS production in the firn." We also added a sentence to this paragraph that explicitly states the need for an organic sulfur source for abiotic COS production before the already existing sentence that suggests marine aerosols could serve as this source (L616-618): "Abiotic COS production also requires an organic sulfur source. Antarctica is far from continental land masses, suggesting marine aerosols could also be the primary source of organic sulfur."

- Given the large uncertainty in the dimethyl sulfide (DMS) contribution to the global ocean COS budget (53 to 680 GgS yr$^{-1}$; Jernigan et al., 2022, https://doi.org/10.1029/2021GL096838), it seems hasty to suggest that atmospheric COS is sensitive to changes in DMS emissions.

The manuscript does not propose that the paleoatmospheric variability of COS reflects changes in DMS emissions, although this is certainly possible within the uncertainties. The sentence on L729-731 makes this clearer: "We cannot quantify how much of the ocean COS source increase results from DMS versus COS and CS$_2$ because of the complexities of ocean production mechanisms of sulfur gases and the uncertainties in their contribution to the atmospheric COS budget." Instead, we argue that a large increase in ocean COS emissions is unlikely to happen without an increase in ocean DMS emissions as stated on L735-736). This reviewer comment seems related to the first specific comment below. We added some additional discussion to clarify and strengthen the argument about inferences related to DMS (L736-740) that are also discussed below.

- Without paleoceanographic evidence, it seems speculative to single out changes in coastal upwelling zones and low-latitude oceans as likely contributors to the deglacial rise in COS. At best this represents one scenario among many possibilities.

Inline with the reviewer's suggestion, we merely suggest coastal upwelling zones are one of the possible contributors to deglacial rise in COS. We discuss multiple possible mechanisms and the relevant sentence (L726-728) comes at the end of the discussion.

- It was not clear why a climate-driven relationship between ssNa$^+$ and COS has to be an anticorrelation. What are the biogeochemical or climatological reasons behind this?

There is no a priori assumption that the climate-driven relationship has to be an anticorrelation. The data analyses reveal an anticorrelation during the glacial period and a positive one during the Holocene. We opted not to speculate on the possible causes in this manuscript because the causes of ssNa variability in the glacial sections of the South Pole ice core have not yet made it into the literature.

**Specific comments**
- L24–25: Seems speculative. The results may allow us to infer a likely increase in total COS emissions, but not in the emissions of each precursor.

This is an interesting point. From our perspective, the least speculative option is to suggest that emissions of all three gases increased. Of course, the details of where exactly most of the emission changes can come from and how much from each gas requires a detailed modeling effort, beyond the scope of this manuscript. We added a sentence to this effect on L739-740 in the revised version.

All ocean sulfur gas emissions ultimately stem from organic life in the ocean. Even though net emissions of COS and CS2 are geographically decoupled, DMS emissions overlap with net emissions of both gases, and all three gases are emitted at coastal upwelling zones. Additionally, when all three gases increase, relatively smaller increases are required for each gas to achieve higher overall OCS in the atmosphere. If only COS and DMS or CS2 and DMS increased, due to changes in upwelling regimes only in the high latitudes or only in the low latitudes, for example, much larger changes in gas emissions would be required within that region to make up for the missing increase from the other region.

There is paleoclimate evidence that upwelling regimes in both high and low latitudes were different during the LGM, and sea level rise during the deglaciation (100-120 m) had dramatic impact on coastal regions and processes close to the continental shelves. We provide multiple references in the manuscript in the relevant paragraph (L720-735) which we supported with a few more in the revised version.

- L33: "COS and CS$_2$ are produced primarily by photochemical reactions" - COS and CS$_2$ are also produced in the dark. See Lennartz et al. (2019) *Ocean Sci.* (https://doi.org/10.5194/os-15-1071-2019) and Modiri Gharehveran and Shah (2018) *ES&T* (https://doi.org/10.1021/acs.est.8b01618).

The evidence for dark production of CS2 in the ocean is limited but we modified the text to reflect this possible production pathway for CS2.

- L35: "Warmer waters can act as a seasonal sink due to temperature dependent loss to hydrolysis" - But both COS and CS$_2$ are less soluble in warmer waters (De Bruyn et al., 1995, https://doi.org/10.1029/95JD00217), and wouldn't this lead to more outgassing from the ocean?

This is addressed in Appendix E (previously Appendix B). Briefly, the hydrolysis loss rate of COS is more sensitive to a unit temperature change than its solubility. Observations leave little room for doubt that COS emissions are low and CS2 emissions are high at low latitudes (Lennartz et al., Earth Syst. Sci. Data, 12, 591–609, 2020). We added this citation to the manuscript.

- L52: It would be helpful to add a brief note on how the bubble–clathrate transition zone (BCTZ) affects the preservation of ancient air in ice cores for those who are not familiar with the BCTZ.

Dry extraction measurements from the BCTZ display large negative biases (L52 and L82-83). The readers are referred to a citation provided in the text (Aydin et al., JGR, 2016) for more detailed information.

- L80: How were the extraction efficiencies of the wet and dry methods characterized? If these methods have been previously examined, a citation would help.

Wet extraction efficiency is close to 100% because solubility of air in water is negligibly small in the context of determining extraction efficiency; Nicewonger et al. (2020) estimate that about 0.7% of the air is left dissolved in the melt water in our extraction vessels. The dry extraction efficiency can be estimated from comparison of total air content data, which is estimated reasonably well by wet extraction measurements from the same core like what was done here, or by comparisons with total air content data from the same core which may be available from a different lab   previously by Aydin et al. (2016). We added citations to the text.

- L112, Eq. (1): Is α the ratio of aqueous concentration divided by gaseous concentration?

Yes. It is similar to the dimensionless Henry's Law solubility except it is the value of effective solubility instead of saturation value. We added this information to the manuscript and changed this coefficient to *h* to avoid confusion ().

- L165, Eq. (8): Is α here the same as that in Eq. (1)? If not, please use a different symbol.

We changed Eq. (1) and kept Eq. (8) the same.

- L191–192: The numbers of effective sample sizes should be reported properly as 4900, 2500, etc.

Done.

- L197: "The 2σ uncertainty ranges shown in the figures represent" - Which figures?

Figs. 4b and 5 (L221).

- L215–L292: This section seems to belong to the supplementary material or methods. I would distill a few key messages only to put in the results section.

For a different audience, this information is necessary to include here.

- L215: "COS was measured at the over the length of ..." - Check typos here.

Done.

- L220: Fig. A1a shows the amount of gas extracted, not gas extraction efficiency per se.

We modified the sentence to reflect this nuance (L242).

- L222–232: This paragraph left me wondering: are the spikes real or not?

We cannot say with confidence and state "it seems unlikely that the COS mixing ratio in the glacial atmosphere varied abruptly at the magnitude and frequency of these spikes (L229-230)."

- L255: "This reversal occurs at a depth where ice impurity concentrations are increasing steeply (Fig. 1b)." - I believe here you intended to reference Fig. 1c.

Yes, corrected.

- L256–264: This paragraph seems to belong to the methods.

This paragraph is necessary to understand the following paragraph.

- L265: "significant" -> "statistically significant"?

Thanks, we changed the sentence to read "statistically significant."

- L266–267: "The slope is stronger ... with ssNa" - If the mechanism by which COS is produced from non-sea-salt aerosols differs from sea salt aerosol-caused COS production, can the slopes be compared on the same scale?

The slopes cannot but the significance can. We changed the sentence to reflect this (L289).

- L269–271: Skimming these sentences the first time, I was confused how an $R^2$ value of 0.04 could serve as the basis for the correction. The next time I realized that this was not the correlation between excess COS (the noise, which is unknown) and ssNa$^+$, but that between the total COS (signal + noise) and ssNa$^+$. You might want to add a brief note somewhere to get this point across more effectively.

Thanks, we modified the sentence to include this information (L287-289).

- L276: "In fact, the presence of spikes is a contributing factor to the low $R^2$ of the correlations despite the high significance" - You might want to point the readers to scenario G2 here to show that the issue of spikes has been taken into account.

Good idea. Done (L298).

- L293–304: This seems to belong to the methods.

This paragraph has been moved to the methods with minor modifications (L171-183).

- L303–304: It was not obvious to me how these analysis scenarios supported the idea that "the relationship between ssNa and COS is driven primarily by millennial scale variability."

We show higher frequency variability do not correlate and lower frequencies do as demonstrated by higher significance of correlation for more smoothed records.

- L317–318: Why was it necessary to scale up the errors? Section 2.4 asks me to go to section 3.1, but section 3.1 refers me back to section 2.4.

This is mostly done out of abundance of caution. The errors are scaled up to test the sensitivity of results to possible underestimation (negative bias) of errors. We included a sentence to this effect on L189-190.

- L334–342: Why are measurements of the glacial period and the Holocene corrected separately? Do you expect the relationship between ssNa$^+$ and excess COS to differ between the last glacial period and the Holocene?

There are reasons they could be different because during the deglaciation ssNa sources regions and the concentration of whatever else is co-deposited can change as well as accumulation rate. In the end, the results show a minor difference, implying a correction to the entire record using only the glacial slope does not change the outcome significantly enough to impact the interpretation (Fig. 4b).

- L345: "150 ppt" and "90% lower" - Check the numbers. It's not like that the Holocene has a COS level at 1500 ppt. Also the referenced figure does not seem to tell this information.

Thanks for catching this. The sentence was initially written in a reverse sense (i.e. 90% rise from the glacial period). We revised the sentence to refer to the ppt difference instead (L385).

- L370–376: The point of these three analyses is lost on me. How do the regressions of smoothed ssNa$^+$ onto unsmoothed ssNa$^+$ support the validity of climate-driven same-age anticorrelation between ssNa$^+$ and COS?

The key component of the analysis is that, in all three control scenarios, one of ssNa records is on the gas chronology while the other ssNa record is on ice chronology. We realize this may be confusing for readers who are not ice core scientists, but this is a valid way to test whether the applied same-depth correction results in, or how much it contributes to, the same-age correlation. These tests are akin to testing the autocorrelation of the ssNa record at a given lag, but instead of a fixed lag, the delta-age for the ice core is used. We modified the associated text in the manuscript to better explain what was done and the meaning of it (L1106-1112). Note that this entire section has been moved to Appendix C in the revised manuscript.

- L454: "COS production in the firn is approximated by an advective-diffusive model of the South Pole firn" - Has this correction been applied to the present study?

The model is presented as a proof of concept. No correction has been made to the measurements using the firn model. We reworded this sentence and added additional clarifying text in the revised version (L610-614).

- L487: "It is possible atmospheric COS is sensitive to changes in ocean DMS emissions modulated by winter sea ice" - According to Lana et al. (2011) climatology, DMS emission hotspots seem to lie beyond the sea-ice covered regions of the Southern Ocean.

Seasonal sea ice is different than the climatological means used in DMS inventories in that dissipation of winter sea ice commonly leads to enhanced DMS emissions, although the reasons remain an open scientific question. Two citations were included in the manuscript (Curran and Jones, 2000; King et al., 2019 L640). There are others suggesting different mechanisms, including one cited by Lana et al. (2011): Trevena, A. J., and G. B. Jones (2006), Dimethylsulphide and dimethylsulphoniopropionate in Antarctic sea ice and their release during sea ice melting, Mar. Chem., 98(2–4), 210–222, doi:10.1016/j.marchem.2005.09.005.

- Fig. 1: For wet extraction results, maybe show only the solubility-corrected values (red line in 1a)? Captions seem misplaced for panels **b** and **c**.

The motivation here is to present the measured wet extraction results so they can see for themselves that the solubility correction itself does not introduce trends to the record.

- Fig. 2: I don't see the "black error bars" described in the caption.

Thanks for catching this. Black error bars were used in a previous version of this figure. Corrected the caption to refer to magenta circles.

- Fig. 4a: Isn't this panel a repeat of Fig. 1a, but with error bars? It may be removed because it adds little new information.

One of the purposes of Fig. 1a is to display the impact of the solubility correction. We opted not to use error bars in that figure for purposes of clarity. An errorbar plot version of the measurements are shown again in Fig4a for two reasons: 1) It provides an immediate comparison with the corrected records under different scenarios shown in Fig. 4b, 2) it displays the errorbars used in the Bayesian analysis that underpin the 2 sigma uncertainty bands of the corrected records in Fig. 4b.

- Fig. 5: Could be combined with Fig. 4 for the comparison.

This is a good idea for comparison purposes, but we suspect the figure will get too large with four panels and the accompanying caption to fit in one page.

- Fig. 6: Seems like a supplementary figure to me.

We anticipate this to be a compelling figure for ice core scientists, specifically for firn air modelers, who are interested in how gas production in the firn can influence ice core gas records.

We thank Dr Wu Sun for his insightful and detailed review of the manuscript and hope that our responses are sufficient to make the best use of his effort.

---

## Author Comment (AC2)

The reviewer comments are in black font and our responses are in blue.

**Reply to reviewer #2**

Summary:
Aydin et al present a new 52kyr record of COS from the South Pole ice core (SPC14). The record was generated using both a dry-extraction (most of the samples) and a wet-extraction technique. An empirical solubility correction is applied to the wet-extracted samples based on comparison with dry-extraction data over the Holocene, where analytical artifacts are thought to be insignificant for the dry technique. Prior work has shown that COS appears to slowly degrade in glacial ice, with the hypothesized mechanism being hydrolysis. The authors argue that SPC14 is too cold for this process to be significant, providing an important advantage in terms of COS preservation. However, the authors find an unrealistically large amount of COS variability in the part of the record from the last glacial period. They observe a very weak but statistically significant correlation between COS and sea-salt sodium (ssNa) concentrations. The authors propose that there is a mechanism in the firn (and firn only) that somehow produces excess COS of non-atmospheric origin. On the basis of the observed correlation with ssNa, the authors develop and apply a correction for excess COS for SPC14. They also apply a version of this correction to COS data from two other ice cores: WAIS Divide and Taylor Dome. The resulting 52kyr record suggests very large changes in atmospheric COS, with 2 – 4 times greater COS in the Holocene than during the last glacial maximum (LGM). This large change is interpreted as an increase in the oceanic sources of COS during the deglaciation.
One point we would like to respond to in the summary is the passing comment about the statistics. When it comes to the validity and the accuracy of the correction, we note very high statistical significance of the relationship and its robustness across various data analysis scenarios.

Major comments:
COS is present in air in very low concentrations (ppt level), and thus is extremely challenging to measure in glacial ice. The authors are therefore to be commended on the very demanding analytical effort involved in producing this record. This is an interesting record, and I think should be published if the authors are able to address the points below, but I find this record very challenging to interpret and have some concerns and recommendations, as follows.
I am not convinced by the authors' arguments for a ssNa-based COS correction. The argument that excess COS is produced only in the firn does not make sense to me. The authors effectively propose a mechanism that "burns out" fast (so that there is nothing below the firn zone). Such a "fast burn" mechanism would be expected to be most intense near the surface, but here ventilation to atmosphere would remove any excess COS that is produced. Trapped air below the firn layer would be much more sensitive to any excess production because there is less air and because this air can't exchange with overlying air.

The manuscript includes results from a firn model that demonstrates the impacts of production in the firn. It is good to see the reviewer's expectations overlap with what we find in our modeling experiment. We do not perceive a direct challenge to the evidence presented for presence of excess COS (i.e., the disagreement between measured COS levels in different ice

cores and the same-depth correlation with impurities). We therefore assume that the reviewer's skepticism about production in the firn is due to the fact that reasons supporting a firn production mechanism were not presented in sufficient detail. We revised the manuscript to expand the arguments supporting production in the firn to emphasize the points outlined below (L632-640).

We propose a production mechanism largely confined to the firn, but strictly speaking, the production does not have to stop immediately below the close-off depth, rather any production that happens deeper in the ice sheet has little to no impact on the record. One exception to this might be the spikes that are unlikely to be atmospheric in origin as discussed in the manuscript (L667-670). We demonstrate with multiple analysis scenarios that the spikes do not impact the interpretation offered in the manuscript.

A production process confined primarily to the firn is sensible because all chemical reactions require substrates which run out over time, lowering the probability of reactions as the ice ages. Emerging laser ablation-based research shows that solid and dissolved phase ice impurities, e.g. sea salt aerosols and trace metals that can act as reaction substrates and catalysts for production of COS, can migrate to different locations in the ice via post-depositional processing (Stoll et al., The new frontier of microstructural impurity research in polar ice, Annals of Glaciology, 2023; this reference added to the manuscript), further lowering the probability of complex reactions deeper in the ice sheet.

As for the reaction rates, the firnification at the South Pole happens over a 1000 year time scale (2000 years during the glacial period); even at fast accumulation sites, the firn-ice transition can take hundreds of years. This is plenty of time for even slow chemical reactions to run their course and substrates to run out or get physically separated from each other.

Any production process that happens in the ice sheet has to include a firn component, with the only possible exceptions being very deep, warm and wet ice with incorporated bedrock materials. If the production continues over very long time horizons (e.g. 10,000-100,000 years) such that what is produced in the ice far exceeds what is produced in the firn, we would expect the excess gas to increase with depth (over time). There is no such evidence in our record.

The ssNa-based correction is very large (up to ≈70% of the measured values for some samples) and results in a record that shows 2 – 4 times lower COS at LGM than at the Holocene, and the authors argue that this must be mainly source-driven. Sources are mainly linked to ocean microbiota, and I find it difficult to accept that oceanic sources could have declined this much (the surface ocean was quite productive during the LGM).
In the absence of any comments on the accuracy of the correction and the related uncertainty estimates, we fail to see the merit in the comment about the magnitude of the correction.

Our argument about the change being primarily source driven is strong, which we assume is agreed by the reviewer given that they do not dispute any specific aspect. The levels of ocean productivity during the LGM is an open science question and we present a well-referenced discussion supporting our interpretation (L693-737).

The fairly good (although far from perfect) agreement among three ice cores after the ssNa correction is encouraging, but the ssNa correction may or may not be transferrable between sites.

The South Pole record is the most detailed and complete COS record and stands on its own. The other ice cores provide enough supporting information for identification and corrections of the excess COS. In fact, one of the strong piece of evidence for the presence of excess COS comes from the comparison of measurements from different sites. The measured COS levels from the different ice cores do not agree, most significantly during the deglaciation, which we can only explain by a production process linked to impurities.

Considering the above, I would recommend presenting the ssNa correction and the resulting temporal trend as speculative and one of possible scenarios (the other obvious scenario being no ssNa correction), and more clearly emphasizing how uncertain the interpretation of the measurements is, both in terms of the resulting reconstruction and implications for COS budget, which Table 1 shows to be very uncertain even today.

An interpretation of the uncorrected WAIS Divide record was published by Aydin et al. (2016), suggesting an increase in GPP might have caused the decline during mid through the late deglaciation. The uncorrected South Pole record did not agree with the WAIS Divide record during this period and indicated a different interpretation was in order. We revised the manuscript to emphasize why the previous interpretation based on the WAIS Divide ice core is not valid (in section 3.2 about comparisons with other ice core records) and further clarify why a correction for excess COS is required before any interpretation can be attempted (L430-435).

We reiterate our confidence about the completeness of our uncertainty analyses and the uncertainty estimates presented for the corrected COS record. The interpretations are limited to the likely ranges provided by the 2 sigma uncertainty estimates and do not include any over interpretations. The reviewer does not comment on any specific aspect of the interpretation for us to address.

Regarding the uncertainties in the COS budget, we offer an interpretation of the record based on the contemporary understanding of COS biogeochemistry. COS literature has been nearly unanimous for about 40 years in identifying ocean emissions as the most important natural source of COS. There is ongoing debate over how much each gas contributes but the impacts of this uncertainty on our interpretation is limited. The discussions we added based on one of reviewer #1 comments also include the possible impact on the interpretation should future research show that one or more of the ocean COS emission components (e.g. emissions from low latitudes versus emissions from high latitudes) is clearly the dominant atmospheric source. This could mean the changes in the COS record reflect more regional changes than global (L748-750).

Minor comments:
I think the manuscript could benefit from a stronger explanation for the motivation for this study, which clearly involved a great deal of effort. It seems that direct radiative forcing due to COS is negligibly small. The authors mention its link with DMS (which has a larger forcing), but

DMS appears to be a relatively smaller source of COS. COS is removed by terrestrial plant uptake – could this be a stronger motivation for the record, as a possible proxy for terrestrial biospheric productivity?

The motivation of this study is to explore what can be learned about past changes in the Earth Systems from a paleoatmospheric record of COS. The interpretation is worked out after the measurements are made and the data analyses are completed. In essence, the interpretation goes where the data takes it. Had we thought there was any possible inference to draw from this record about past GPP variability, it would have been in the manuscript.

Section 2.2:

The temperature-dependent expression for COS solubility should be given, and the actual solubility value used should be stated, and compared to the value for air. What fraction of COS is typically in the meltwater?

We deploy an empirical method in the manuscript and do not explicitly use the solubility of COS. The temperature dependent solubility equations from Sander et al. (2015) for COS and air are provided by the citation in the paper (supplemental Table 6 of Nicewonger et al., 2020). We revised the manuscript to include a more explicit citation to this previous work. The solubility correction is a factor of 1.2 on average (L149), meaning about 20% of the COS in ice core air is left in the melt water. The full distribution of corrections is shown in Fig. A3. Nicewonger et al. (2020) estimated that roughly 25% of COS would be left dissolved in the melt water. In other words, melt water is slightly undersaturated at the completion of the wet extraction. This amount varies from sample to sample (Fig. A3).

Line 149: "above" and "below" ◊ "shallower than" and "deeper than" would be less ambiguous here. "above" could mean "greater than"

Thanks, changed as suggested.

Line 215: "COS was measured over the length of the SPC14 ice core…"

Thanks, corrected.

Line 252: Do you mean "ice from the last glacial period"? All of the ice core is "glacial ice". Please edit to clarify.

Changed to glacial period ice.

Line 254: Fig 1c does not show COS during time interval being discussed

Changed to Fig. 1b.

We thank the reviewer for their concerns and suggestions.

---

## Author Response (AR2)

Dear Editor and Reviewers,

We are grateful to the attention to detail in the manuscript feedback. Below are our comments. The reviewer comments are in black font and our responses are in blue.

**Reply to Wu Sun (reviewer #1)**

In this work, Aydin et al. painstakingly recovered a 52,500-year record of atmospheric carbonyl sulfide from a South Pole ice core, with great care and attention devoted to correcting for the post-depositional COS production from sea salt aerosols and other artifacts during extraction. The resulting data set is a valuable contribution to the atmospheric history of carbonyl sulfide and will enable the climate and ecosystem modeling community to better understand biospheric changes since the last ice age.

Despite its scientific significance, the organization of the manuscript may hinder its key findings from being grasped by a broad audience. For this reason, I suggest clarifying the methods, especially the rationale behind every correction and sensitivity test, and streamlining the presentation of the main messages to strengthen the work. Below I list a few high-level issues followed by specific line-by-line comments.

My issues with the methods are:

- For the calculation of ssNa$^+$ (Sect. 2.3), please check the denominator in the right hand side of Eq. (6). My derivation says it should be "$1 - R_m/R_t$" in the denominator (see attached slides). In other words, there is no reason for Eqs. (6) and (7) to have different denominators.

The reviewer is correct. Both denominators should display a subtraction of Rm/Rt from 1. This is now corrected. This change results in a linear 4% increase in the calculated ssNa and 4% change in the slope values involving regressions vs. ssNa. This does not result in any change in the data analysis results, including the corrections because the change in the ssNa is compensated by the same proportional change in the correction slopes. All relevant figures have been updated for this minor change in ssNa.

- For the regression between COS and ssNa$^+$ (Sect. 2.4), it was not clear why errors in the response variable need to be scaled with a multiplier (α in Eq. 8). This multiplier did not appear in Eq. (5).

We included alpha as a multiplicative scaler in eq. 8 to demonstrate the impact of a possible bias in determination of the measurement errors (e.g. consistent underestimation of the measurement errors) on the regression results. The alpha parameter is 1 for three of the scenarios (G1, G2, H1) and its distribution is determined by the Bayesian algorithm in the other scenarios, allowing us to demonstrate the difference between using calculated measurement errors versus what the impact on the results is if the measurement errors are biased low. We included new text in section 2.4 to clarify (L206-207): "Inclusion of the $\alpha$ parameter allows quantification of the sensitivity of the correction slope to possible bias of COS measurement errors."

- It was also unclear how measurement errors in COS and ssNa⁺ (Sect. 2.5) were incorporated into the Bayesian errors-in-variables regression between COS and ssNa⁺ (Sect. 2.4).

The COS errors are directly used as measurement errors (i.e. errors in y, denoted yerr in equations 8, 9, and 10). The ssNa errors are used as xerr in eq. 4. We added text to section 2.4 to clarify (L188-191): "The error estimates for the COS and ssNa measurements are directly incorporated in the errors-in-variables regressions as they substitute for $y_{err}$ and $x_{err}$ in equations 4 and 8, allowing us to propagate the uncertainty in the measurements to the slope of the relationship between ssNa-COS."

- Different scenarios for correcting for ssNa⁺-produced excess COS (Table 1) should be clearly outlined in the methods section, with the rationale explained. It was not until I read the results did I get a hint of why these scenarios were needed and why they were set up like these. The choice of the averaging window size appears ad hoc.

The manuscript is revised to introduce the different scenarios under methods in section 2.4 with a sufficient amount of detail about their purpose (L191-198). Some details of the scenarios are still presented under results in section 3.1 because decoupling some of the descriptions from the results of the scenarios makes the text incomprehensible.

The reviewer is correct that the exact values of the averaging windows are ad hoc. The 100-y averaging window is chosen to be higher than the span of the ice age of a typical sample (about 10 y during the glacial period) and yield a higher statistical significance. The 1100-y window is chosen to match the averaging window of the smoothing applied to the COS record to diminish the influence of the spikes on the regressions. The results are robust with respect to the averaging windows despite the large range explored in data averaging.

- Too many scenarios overlaid on the same plot (e.g., Figs. 4b and 5) make it hard to discern the key information. Consider presenting only scenarios that are most robust and relevant to the interpretations.

The results from all scenarios are displayed together in Fig. 4b because one of points we are trying to get across is the fact that the interpretations rely only on the features that are common to all scenarios: low during the glacial period, lowest during the LGM, and high during the Holocene. In the original version of the manuscript, Fig. 5 included two less scenarios than Fig. 4b. In the revised version, we eliminated three other scenarios (two from glacial and one from the Holocene) from Fig. 5, reducing the number of scenarios displayed in this figure to three. Fig. 5 is now a fairly plain figure.

The results section needs to provide answers to questions raised in the introduction. Currently it is a mix of methods, results, technical arguments, and discussion, making it challenging to navigate. The main issues are:
- Many paragraphs describe the nitty-gritty details of various corrections and make reference to figures in the appendices. Important as they are to ensuring data quality, these are probably not the high-level findings you want the readers to walk away with. Consider moving them to methods or the appendices.
- Descriptions of the correction for ssNa⁺-produced excess COS are scattered throughout the results section, making it difficult to follow. Consider consolidating the main points

about this under a single subsection and offload nonessential details to supplementary materials.

Based on the reviewer comments, what used to be sections 3.2, 3.3, and 3.4 are now moved to the appendices. These sections included many of the additional data analyses including sensitivity tests for G1 and G4 scenarios, testing the validity of the glacial same-age correlation as a climate signal, and statistical tests regarding the COS rise concurrent with the deglaciation. Atmospheric COS variability inferred from different scenarios are still presented under results in section 3.1 with a modified title "Results of different analysis scenarios: Inferred atmospheric COS variability after the ssNa correction." Section 3.1 contains the details of exactly what was done under each analysis scenario.

- It was not clear how the "climate-driven" correlation between $ssNa^+$ and COS was disentangled from the production artifact. Shouldn't the $ssNa^+$COS production correction be applied first before one can make any robust inference of a climate-driven relationship?

By definition, the climate relationship is observed over time (same-age) and the production relationship is observed over same-depth. If the delta-age is large enough, which is the case for the South Pole ice core, separate same-depth and same-age regressions can reveal both the production and climate driven signal as we show in the manuscript with the independent same-depth and same-age analyses for G1 and H1 scenarios. In a typical data analysis approach, untangling the two relationships from each other would commonly be accomplished by correcting for the production first, followed by quantifying and accounting for the same-age relationship, then cycling back to the same-depth correction, and so forth until the successive corrections do not statistically differ from each other. The simultaneous correction algorithm we deploy here in G4 and H2 scenarios allows carrying out this operation in one step within the Bayesian framework. Note that the software used in this analysis is publicly available and the code is made available as supplemental information. The unique aspect of the code used in this analysis is that it does not make use of an iterative process to accomplish this task.

- West Antarctica and Taylor Dome ice cores (L233–243) need to be described in the methods.

We realize that the names of the ice cores (i.e. WAIS Divide and Taylor Dome) were not explicitly mentioned in the introduction paragraph, likely obscuring the fact that the measurements from these ice cores were presented in prior publications. In the revised version, we modified the introduction paragraph to refer to these ice cores by their names (L52-53) where we describe the findings from these two ice cores, citing the relevant references. Further, we revised the manuscript to add an expanded description of the findings from these ice cores and the known limitations (L54-64).

In the results section, we refer to the results from these ice cores again (L265-275) and then again in section 3.2 where the records from different cores are compared with each other (section 3.4). We note in the methods section that the current SPC14 measurements were conducted with the same measurement methods implemented for these older ice cores (L83-85).

With respect to the interpretations of the data, there are a few issues:

- The measurements presented here only show ssNa$^+$ but say nothing about what the anions in sea salts are. Thus, it is unclear whether sea salt aerosols contain *organic* sulfur compounds and the analogy between COS dark production in the ocean and that in ice (L462–L476) does not seem to be empirically supported. Experimental evidence suggests that COS is produced from organic sulfur compounds (Modiri Gharehveran and Shah, 2018, https://doi.org/10.1021/acs.est.8b01618) such as cysteine, methionine, and chromophoric dissolved organic matter (CDOM). In any case, a reduced sulfur precursor is needed to produce COS in the absence of sulfate-reducing microbes.

The most abundant anion in the ice core is Cl(-), which is strongly correlated with Na(+) since Cl(-) is sourced primarily from NaCl. Sulfate is another important anion but is sourced primarily from inorganic sulfur (SO2) with an additional volcanic component and experiences some post-depositional processing. We conducted regression analyses between COS and all major ions measured in the South Pole ice core, including Cl(-) and SO4(2-). The Cl(-) regression results display linear relationships with COS much like Na(+) for the reason stated above. There is no relationship between COS and SO4(2-). We do not include these other regression analyses in the manuscript because they do not provide additional insight. There are no measurements of organic sulfur from the South Pole ice core that can be used in our analyses, and as far as we know, there are no commonly accepted proxies for organic sulfur content in ice cores.

We acknowledge that an organic sulfur source is needed for abiotic COS production, e.g. (L461-464): "The commonly postulated abiotic process involves reactions between carbonyl groups and thiyl radicals derived from organic sulfur (Flöck et al., 1997; Modiri Gharehveran and Shah, 2018; Lennartz et al., 2017; 2019; Pos et al., 1998; Zepp and Andreae, 1994)."

We do not make any claims of full empirical support for the proposed mechanism (L455-456): "In the absence of auxiliary data, we can only speculate on viable mechanisms for the COS production in the firn." To clarify, this sentence has been revised to read "In the absence of direct empirical evidence, we can only speculate on viable mechanisms for the COS production in the firn."

We also added a sentence to this paragraph that explicitly states the need for an organic sulfur source for abiotic COS production before the already existing sentence that suggests marine aerosols could serve as this source (L467): "Abiotic COS production also requires an organic sulfur source. Antarctica is far from continental land masses, suggesting marine aerosols could also be the primary source of organic sulfur."

- Given the large uncertainty in the dimethyl sulfide (DMS) contribution to the global ocean COS budget (53 to 680 GgS yr$^{-1}$; Jernigan et al., 2022, https://doi.org/10.1029/2021GL096838), it seems hasty to suggest that atmospheric COS is sensitive to changes in DMS emissions.

The manuscript does not propose that DMS emissions account for a large portion of paleoatmospheric COS sources, although this is certainly possible within the uncertainties. The sentence on L573-575 makes this clearer: "We cannot quantify how much of the ocean COS source increase results from DMS versus COS and CS$_2$ because of the complexities of ocean

production mechanisms of sulfur gases and the uncertainties in their contribution to the atmospheric COS budget." Instead, we argue that a large increase in ocean COS emissions is unlikely to happen without an increase in ocean DMS emissions. This reviewer comment seems related to the first specific comment below. We added a substantial amount of additional discussion to clarify and strengthen the argument and reasoning behind the inferences related to an increase in the ocean emissions in general, and about DMS in particular (L571-586) that are also discussed below.

- Without paleoceanographic evidence, it seems speculative to single out changes in coastal upwelling zones and low-latitude oceans as likely contributors to the deglacial rise in COS. At best this represents one scenario among many possibilities.

Inline with the reviewer's suggestion, we merely suggest coastal upwelling zones are one of the possible contributors to deglacial rise in COS. We discuss multiple possible mechanisms citing paleoclimate evidence (L558-570) and the relevant sentence comes at the end of this discussion. We changed the adverb in the beginning of this sentence from "independently" to "additionally" to clarify we do not mean to suggest coastal processes alone can cause deglacial COS rise (L567-570): "Additionally, the coastal emissions of COS, $CS_2$, and DMS can increase as a response to the deglacial sea-level rise and the associated expansion of shelves and coastal seas coupled with increased riverine output of organic matter (Jennerjahn, 2012; Lerman et al., 2011; Peltier and Fairbanks, 2006)."

- It was not clear why a climate-driven relationship between $ssNa^+$ and COS has to be an anticorrelation. What are the biogeochemical or climatological reasons behind this?

There is no a priori assumption that the climate-driven relationship has to be an anticorrelation. The data analyses reveal an anticorrelation during the glacial period and a positive one during the Holocene. We opted not to speculate on the possible causes in this manuscript because the causes of ssNa variability in the glacial sections of the South Pole ice core have not yet made it into the literature.

**Specific comments**
- L24–25: Seems speculative. The results may allow us to infer a likely increase in total COS emissions, but not in the emissions of each precursor.

This is an interesting point. From our perspective, the least speculative option is to suggest that emissions of all three gases increased. Of course, the details of where exactly most of the emission changes can come from, and how much from each gas, requires better constraints on the atmospheric budget and a detailed modeling effort, which are beyond the scope of this manuscript. We revised the final sentence of the abstract to convey the uncertainties (L24-26): "A large increase in ocean COS emissions during the deglaciation suggests enhancements in emissions of ocean sulfur gases via processes that involve ocean productivity, although we cannot quantify individual contributions from each gas." We also added a sentence to the conclusions for the same purpose (L619-622): "Better constraints on the atmospheric COS budget, particularly on the specifics of the ocean sources, coupled with a modelling effort are needed to quantitatively partition the necessary emissions increases among different sources and to refine climate implications."

All ocean sulfur gas emissions ultimately stem from organic life in the ocean. Even though net emissions of COS and CS2 are geographically decoupled, DMS emissions overlap with net emissions of both gases, and all three gases are emitted at coastal upwelling zones. Additionally, when all three gases increase, relatively smaller increases are required for each gas to achieve higher overall OCS in the atmosphere. If only COS and DMS or CS2 and DMS increased, due to changes in upwelling regimes only in the high latitudes or only in the low latitudes, for example, much larger changes in gas emissions would be required within that region to make up for the missing increase from the other region.

There is paleoclimate evidence that upwelling regimes in both high and low latitudes were different during the LGM, and sea level rise during the deglaciation (100-120 m) had dramatic impact on coastal regions and processes close to the continental shelves.

We revised the relevant sections to strengthen the relevant discussions (L571-586), which now include the information summarized above.

- L33: "COS and CS$_2$ are produced primarily by photochemical reactions" - COS and CS$_2$ are also produced in the dark. See Lennartz et al. (2019) *Ocean Sci.* (https://doi.org/10.5194/os-15-1071-2019) and Modiri Gharehveran and Shah (2018) *ES&T* (https://doi.org/10.1021/acs.est.8b01618).

In the introduction, we present a brief summary of the COS budget mentioning only the primary mechanisms. A more detailed discussion of ocean sources is introduced later in the text, including a reference to the dark production of COS (L531). The evidence for dark production of CS2 is more limited, but we revised the manuscript to also refer to this possible production mechanism (L537-538).

- L35: "Warmer waters can act as a seasonal sink due to temperature dependent loss to hydrolysis" - But both COS and CS$_2$ are less soluble in warmer waters (De Bruyn et al., 1995, https://doi.org/10.1029/95JD00217), and wouldn't this lead to more outgassing from the ocean?

This is addressed in Appendix E (previously Appendix B). Briefly, the hydrolysis loss rate of COS is more sensitive to a unit temperature change than its solubility. Observations leave little room for doubt that COS emissions are low and CS2 emissions are high at low latitudes (Lennartz et al., Earth Syst. Sci. Data, 12, 591–609, 2020). We added this citation to the manuscript (L577).

- L52: It would be helpful to add a brief note on how the bubble–clathrate transition zone (BCTZ) affects the preservation of ancient air in ice cores for those who are not familiar with the BCTZ.

We added a description of the BCTZ in the introduction (L56-59).

We also revised the methods to explain why dry extraction is less efficient in extracting air from clathrates (L92-94) and why gas fraction can occur between bubbles and clathrates (L95-100) with relevant references.

- L80: How were the extraction efficiencies of the wet and dry methods characterized? If these methods have been previously examined, a citation would help.

Wet extraction efficiency is close to 100% because solubility of air in water is negligibly small in the context of determining extraction efficiency; Nicewonger et al. (2020) estimate that about 0.7% of the air is left dissolved in the melt water in our extraction vessels (cited on L90). The dry extraction efficiency can be estimated from comparison of total air content data, which is estimated reasonably well by wet extraction measurements from the same core like what was done here, or by comparisons with total air content data from the same core which may be available from a different lab as noted previously by Aydin et al. (2016).

- L112, Eq. (1): Is α the ratio of aqueous concentration divided by gaseous concentration?

Yes. It is similar to the dimensionless Henry's Law solubility except it is the value of effective solubility instead of saturation value. We added this information to the manuscript and changed this coefficient to $h$ to avoid confusion (L130-131).

- L165, Eq. (8): Is α here the same as that in Eq. (1)? If not, please use a different symbol.

We changed Eq. (1) and kept Eq. (8) the same.

- L191–192: The numbers of effective sample sizes should be reported properly as 4900, 2500, etc.

Done.

- L197: "The 2σ uncertainty ranges shown in the figures represent" - Which figures?

Figs. 4b and 5 (L227).

- L215–L292: This section seems to belong to the supplementary material or methods. I would distill a few key messages only to put in the results section.

For a different audience, this information is necessary to include here.

- L215: "COS was measured at the over the length of ..." - Check typos here.

Deleted "at the".

- L220: Fig. A1a shows the amount of gas extracted, not gas extraction efficiency per se.

We modified the sentence to reflect this nuance (L251-252).

- L222–232: This paragraph left me wondering: are the spikes real or not?

We cannot say with confidence and state (L263-265) "it seems unlikely that the COS mixing ratio in the glacial atmosphere varied abruptly at the magnitude and frequency of these spikes."

- L255: "This reversal occurs at a depth where ice impurity concentrations are increasing steeply (Fig. 1b)." - I believe here you intended to reference Fig. 1c.

Thanks for catching this. Corrected (L288).

- L256–264: This paragraph seems to belong to the methods.

We deem this paragraph necessary for understanding the following paragraph.

- L265: "significant" -> "statistically significant"?

Thanks, we changed the sentence (L299) to read "statistically significant."

- L266–267: "The slope is stronger ... with ssNa" - If the mechanism by which COS is produced from non-sea-salt aerosols differs from sea salt aerosol-caused COS production, can the slopes be compared on the same scale?

The slopes cannot but the significance can. We changed the sentence to reflect this (L301).

- L269–271: Skimming these sentences the first time, I was confused how an $R^2$ value of 0.04 could serve as the basis for the correction. The next time I realized that this was not the correlation between excess COS (the noise, which is unknown) and $ssNa^+$, but that between the total COS (signal + noise) and $ssNa^+$. You might want to add a brief note somewhere to get this point across more effectively.

Thanks, we modified the sentence to include this information (L304-307).

- L276: "In fact, the presence of spikes is a contributing factor to the low $R^2$ of the correlations despite the high significance" - You might want to point the readers to scenario G2 here to show that the issue of spikes has been taken into account.

Good idea. Done (L314).

- L293–304: This seems to belong to the methods.

This paragraph has been moved to the methods with minor modifications (L187-206).

- L303–304: It was not obvious to me how these analysis scenarios supported the idea that "the relationship between ssNa and COS is driven primarily by millennial scale variability."

We show higher frequency variability do not correlate and lower frequencies do as demonstrated by higher significance of correlation for more smoothed records.

- L317–318: Why was it necessary to scale up the errors? Section 2.4 asks me to go to section 3.1, but section 3.1 refers me back to section 2.4.

This is mostly done out of abundance of caution. The errors are scaled with a multiplier to test the sensitivity of results to possible biases in determination of errors. As previously noted under responses to general comments, we included revised text to clarify (L206-207).

- L334–342: Why are measurements of the glacial period and the Holocene corrected separately? Do you expect the relationship between $ssNa^+$ and excess COS to differ between the last glacial period and the Holocene?

There are reasons they could be different because during the deglaciation ssNa sources regions and the concentration of whatever else is co-deposited can change as well as accumulation rate. In the end, the results show a minor difference, implying a correction to the entire record using only the glacial slope does not change the outcome significantly enough to impact the interpretation (Fig. 4b).

- L345: "150 ppt" and "90% lower" - Check the numbers. It's not like that the Holocene has a COS level at 1500 ppt. Also the referenced figure does not seem to tell this information.

Thanks for catching this. The sentence was initially written in a reverse sense (i.e. 90% rise from the glacial period). We revised the sentence to refer to the ppt difference instead (L371-372), which is easier to see in Fig. 3f.

- L370–376: The point of these three analyses is lost on me. How do the regressions of smoothed ssNa$^+$ onto unsmoothed ssNa$^+$ support the validity of climate-driven same-age anticorrelation between ssNa$^+$ and COS?

Note that this section has been moved to Appendix C in the revised manuscript while the scenarios are introduced earlier in the methods section (L191-206) following reviewer's suggestions. The key component of the analysis is that, in all three control scenarios, one of ssNa records is on the gas chronology while the other ssNa record is on ice chronology. We realize this may be confusing for readers who are not ice core scientists, but this is a valid way to test whether the applied same-depth correction results in, or how much it contributes to, the same-age correlation. These tests are akin to testing the autocorrelation of the ssNa record at a given lag, but instead of a fixed lag, the delta-age for the ice core is used. We modified the associated text in the manuscript, providing this information (L972-973).

- L454: "COS production in the firn is approximated by an advective-diffusive model of the South Pole firn" - Has this correction been applied to the present study?

The model is presented as a proof of concept. No correction has been made to the measurements using the firn model. We reworded this sentence to clarify (L446-447): " To demonstrate the viability of the proposed production process, COS production in the firn is simulated within an advective-diffusive model of the South Pole firn (Aydin et al., 2020)."

- L487: "It is possible atmospheric COS is sensitive to changes in ocean DMS emissions modulated by winter sea ice" - According to Lana et al. (2011) climatology, DMS emission hotspots seem to lie beyond the sea-ice covered regions of the Southern Ocean.

Seasonal sea ice is different than the climatological means used in DMS inventories in that dissipation of winter sea ice commonly leads to enhanced DMS emissions, although the reasons remain an open scientific question. Two citations were included in the manuscript (Curran and Jones, 2000; King et al., 2019 L640). There are others suggesting different mechanisms, including one cited by Lana et al. (2011): Trevena, A. J., and G. B. Jones (2006), Dimethylsulphide and dimethylsulphoniopropionate in Antarctic sea ice and their release during sea ice melting, Mar. Chem., 98(2–4), 210–222, doi:10.1016/j.marchem.2005.09.005.

- Fig. 1: For wet extraction results, maybe show only the solubility-corrected values (red line in 1a)? Captions seem misplaced for panels **b** and **c**.

The motivation here is to present the measured wet extraction results so the readers can see for themselves that the solubility correction itself does not introduce trends to the record. Thanks for catching that the captions were mislabeled. Corrected now.

- Fig. 2: I don't see the "black error bars" described in the caption.

Thanks for catching this. Black error bars were used in a previous version of this figure. Corrected the caption to refer to magenta circles.

- Fig. 4a: Isn't this panel a repeat of Fig. 1a, but with error bars? It may be removed because it adds little new information.

One of the purposes of Fig. 1a is to display the impact of the solubility correction. We opted not to use error bars in that figure for purposes of clarity. An errorbar plot version of the measurements are shown again in Fig4a for two reasons: 1) It provides an immediate comparison with the corrected records under different scenarios shown in Fig. 4b, 2) it displays the errorbars used in the Bayesian analysis that underpin the 2 sigma uncertainty bands of the corrected records in Fig. 4b.

- Fig. 5: Could be combined with Fig. 4 for the comparison.

This is a good idea for comparison purposes, but we suspect the figure will get too large with four panels and the accompanying caption to fit in one page.

- Fig. 6: Seems like a supplementary figure to me.

We anticipate this to be a compelling figure for ice core scientists, specifically for firn air modelers, who are interested in how gas production in the firn can influence ice core gas records.

We thank Dr Wu Sun for his insightful and detailed review of the manuscript and hope that our responses are sufficient to make the best use of his effort.

**Reply to reviewer #2**

Summary:
Aydin et al present a new 52kyr record of COS from the South Pole ice core (SPC14). The record was generated using both a dry-extraction (most of the samples) and a wet-extraction technique. An empirical solubility correction is applied to the wet-extracted samples based on comparison with dry-extraction data over the Holocene, where analytical artifacts are thought to be insignificant for the dry technique. Prior work has shown that COS appears to slowly degrade in glacial ice, with the hypothesized mechanism being hydrolysis. The authors argue that SPC14 is too cold for this process to be significant, providing an important advantage in terms of COS preservation. However, the authors find an unrealistically large amount of COS variability in the part of the record from the last glacial period. They observe a very weak but statistically significant correlation between COS and sea-salt sodium (ssNa) concentrations. The authors propose that there is a mechanism in the firn (and firn only) that somehow produces excess COS of non-atmospheric origin. On the basis of the observed correlation with ssNa, the authors develop and apply a correction for excess COS for SPC14. They also apply a version of this correction to COS data from two other ice cores: WAIS Divide and Taylor Dome. The resulting 52kyr record suggests very large changes in atmospheric COS, with 2 – 4 times greater COS in the Holocene than during the last glacial maximum (LGM). This large change is interpreted as an increase in the oceanic sources of COS during the deglaciation.
One point we would like to respond to in the summary is the passing comment about the statistics. When it comes to the validity and the accuracy of the correction, we note very high statistical significance of the relationship and its robustness across various data analysis scenarios.

Major comments:
COS is present in air in very low concentrations (ppt level), and thus is extremely challenging to measure in glacial ice. The authors are therefore to be commended on the very demanding analytical effort involved in producing this record. This is an interesting record, and I think should be published if the authors are able to address the points below, but I find this record very challenging to interpret and have some concerns and recommendations, as follows.
I am not convinced by the authors' arguments for a ssNa-based COS correction. The argument that excess COS is produced only in the firn does not make sense to me. The authors effectively propose a mechanism that "burns out" fast (so that there is nothing below the firn zone). Such a "fast burn" mechanism would be expected to be most intense near the surface, but here ventilation to atmosphere would remove any excess COS that is produced. Trapped air below the firn layer would be much more sensitive to any excess production because there is less air and because this air can't exchange with overlying air.

The manuscript includes results from a firn model that demonstrates the impacts of production in the firn. It is good to see the reviewer's expectations overlap with what we find in our modeling experiment. We do not perceive a direct challenge to the evidence presented for presence of excess COS (i.e., the disagreement between measured COS levels in different ice cores and the same-depth correlation with impurities). We therefore assume that the reviewer's skepticism about production in the firn is due to the fact that the reasons supporting

a firn production mechanism were not presented in sufficient detail. We revised the manuscript to expand the arguments supporting production in the firn (L422-445) to emphasize the points outlined below.

We propose a production mechanism largely confined to the firn, but strictly speaking, the production does not have to stop immediately below the close-off depth, rather any production that happens deeper in the ice sheet has little to no impact on the record. One exception to this might be the spikes that are unlikely to be atmospheric in origin as discussed in the manuscript (L261-264). We demonstrate with multiple analysis scenarios that the spikes do not impact the interpretation offered in the manuscript.

A production process confined primarily to the firn is sensible because all chemical reactions require substrates which run out over time, lowering the probability of reactions as the ice ages. Emerging laser ablation-based research shows that solid and dissolved phase ice impurities, e.g. sea salt aerosols and trace metals that can act as reaction substrates and catalysts for production of COS, can migrate to different locations in the ice via post-depositional processing (Stoll et al., The new frontier of microstructural impurity research in polar ice, Annals of Glaciology, 2023), further lowering the probability of complex reactions deeper in the ice sheet. This citations has been added to the manuscript (L444).

As for the reaction rates, the firnification at the South Pole happens over a 1000 year time scale (2000 years during the glacial period); even at fast accumulation sites, the firn-ice transition can take hundreds of years. This is plenty of time for even slow chemical reactions to run their course and substrates to run out or get physically separated from each other.

Any production process that happens in the ice sheet has to include a firn component, with the only possible exceptions being very deep, warm and wet ice with incorporated bedrock materials. If the production continues over very long time horizons (e.g. 10,000-100,000 years) such that what is produced in the ice far exceeds what is produced in the firn, we would expect the excess gas to increase with depth (over time). There is no such evidence in our record.

The ssNa-based correction is very large (up to ≈70% of the measured values for some samples) and results in a record that shows 2 – 4 times lower COS at LGM than at the Holocene, and the authors argue that this must be mainly source-driven. Sources are mainly linked to ocean microbiota, and I find it difficult to accept that oceanic sources could have declined this much (the surface ocean was quite productive during the LGM).
In the absence of any comments on the accuracy of the correction and the related uncertainty estimates, we fail to see the reasoning behind the comment about the magnitude of the correction.

Our argument about the change being primarily source driven is strong, which we assume is agreed by the reviewer given that they do not dispute any specific aspect. The state of ocean productivity during the LGM is an open science question and we present a well-referenced discussion supporting our interpretation (L558-594).

The fairly good (although far from perfect) agreement among three ice cores after the ssNa correction is encouraging, but the ssNa correction may or may not be transferrable between sites.

The South Pole record is the most detailed and complete COS record and stands on its own. The other ice cores provide enough supporting information for identification and corrections of the excess COS. In fact, one of the strong piece of evidence for the presence of excess COS comes from the comparison of measurements from different sites. The measured COS levels from the different ice cores do not agree, most significantly during the deglaciation, which we can only explain by a production process linked to impurities.

Considering the above, I would recommend presenting the ssNa correction and the resulting temporal trend as speculative and one of possible scenarios (the other obvious scenario being no ssNa correction), and more clearly emphasizing how uncertain the interpretation of the measurements is, both in terms of the resulting reconstruction and implications for COS budget, which Table 1 shows to be very uncertain even today.

An interpretation of the uncorrected WAIS Divide record was published by Aydin et al. (2016), suggesting an increase in GPP might have caused the decline during mid through the late deglaciation. The uncorrected South Pole record did not agree with the WAIS Divide record during this period and indicated a different interpretation was in order. We revised the manuscript to emphasize why the previous interpretation based on the WAIS Divide ice core is not valid (in section 3) and further clarify why a correction for excess COS is required before any interpretation can be attempted (L283-289).

We reiterate our confidence about the completeness of our uncertainty analyses and the uncertainty estimates presented for the corrected COS record. The interpretations are limited to the likely ranges provided by the 2 sigma uncertainty estimates and do not include any over interpretations. The reviewer does not comment on any specific aspect of the interpretation for us to address.

Regarding the uncertainties in the COS budget, we offer an interpretation of the record based on the contemporary understanding of COS biogeochemistry. COS literature has been nearly unanimous for about 40 years in identifying ocean emissions as the most important natural source of COS. There is ongoing debate over how much each gas contributes but the impacts of this uncertainty on our interpretation is limited. The discussions we added based on one of reviewer #1 comments also include the possible impact on the interpretation should future research show that one or more of the ocean COS emission components (e.g. emissions from low latitudes versus emissions from high latitudes) is clearly the dominant atmospheric source. This could mean the changes in the COS record reflect more regional changes than global (L571-586). We also modified the abstract and added text to the conclusions to better convey the uncertainties in our interpretation and outline future work needed to refine the conclusions (L24-26 and L619-622).

Minor comments:
I think the manuscript could benefit from a stronger explanation for the motivation for this study, which clearly involved a great deal of effort. It seems that direct radiative forcing due to

COS is negligibly small. The authors mention its link with DMS (which has a larger forcing), but DMS appears to be a relatively smaller source of COS. COS is removed by terrestrial plant uptake – could this be a stronger motivation for the record, as a possible proxy for terrestrial biospheric productivity?

The motivation of this study is to explore what can be learned about past changes in the Earth Systems from a paleoatmospheric record of COS. The interpretation is worked out after the measurements are made and the data analyses are completed. In essence, the interpretation goes where the data takes it. Had we thought there was any possible inference to draw from this record about past GPP variability, it would have been in the manuscript.

Section 2.2:
The temperature-dependent expression for COS solubility should be given, and the actual solubility value used should be stated, and compared to the value for air. What fraction of COS is typically in the meltwater?

We deploy an empirical method in the manuscript and do not explicitly use the solubility of COS. The temperature dependent solubility equations from Sander et al. (*Compilation of Henry's law constants (version 4.0) for water as solvent. Atmospheric Chemistry and Physics, 15, 4399–4981,* 2015) for COS and air are provided by the citation in the paper (supplemental Table 6 of Nicewonger et al., 2020 cited on L90 and 125). The solubility correction is a factor of 1.2 on average (L167), meaning about 20% of the COS in ice core air is left in the melt water. The full distribution of corrections is shown in Fig. A3. Nicewonger et al. (2020) estimated that roughly 25% of COS would be left dissolved in the melt water if fully saturated. In other words, melt water is slightly undersaturated at the completion of the wet extraction. This amount varies from sample to sample (Fig. A3).

Line 149: "above" and "below" ◊ "shallower than" and "deeper than" would be less ambiguous here. "above" could mean "greater than"

Thanks, changed as suggested.

Line 215: "COS was measured over the length of the SPC14 ice core…"

Thanks, corrected.

Line 252: Do you mean "ice from the last glacial period"? All of the ice core is "glacial ice". Please edit to clarify.

Thanks, changed to glacial period ice for all instances of such use.

Line 254: Fig 1c does not show COS during time interval being discussed

Changed to Fig. 1b.

We thank the reviewer for their concerns and suggestions.

---

## Author Response (AR3)

The authors have largely rejected the major concerns in my first-round review and have argued for the approaches and interpretations presented in their original manuscript. My main comments for the second round of review are as follows.

Upon reviewing the manuscript again, I am more convinced by the correlation between ssNa and COS at the same depths in the ice core. However, I am still not convinced that applying a correction based on this relationship is appropriate.

We regret that this disagreement persists. We insist that applying the ssNa correction is only appropriate course of action. An atmospheric interpretation without applying the correction would mean ignoring the strong evidence for the presence of COS production in the ice sheet. We will not include an interpretation to the manuscript that we do not believe is true.

The authors have assumed that this correlation is due to in situ production. I think this is one possible explanation, but I don't think it is the only possible explanation.

We cannot identify any plausible alternate explanation in the reviewer comments below.

In favor of the authors' assumption and approach is the observation that glacial-period and deglaciation COS value agreement between different ice cores is overall improved by this correction (with the caveat that the WD and TD ice core data seem even more complicated to interpret because of the hydrolysis correction that also has to be applied). However, there remain parts of the record where the different ice cores disagree (most notably around 19 ka and in the later part of the deglaciation). Further, the high scatter in glacial COS values (which was the main reason in situ production was postulated) does not seem to be improved by the correction in the scenario where there is no multi-point smoothing of the COS (Fig 4b, scenario G1).

The reviewer's statement about the high scatter in glacial period COS values being the main reason why in situ production is postulated is not accurate. The in situ production is postulated primarily as an explanation of the discrepancies between records from different sites, specifically during the last deglaciation, which is the most prominent climate event during the period our record covers. The detailed reasoning is presented on L267-297. Below, we present one sentence from this section (L287-289) to demonstrate this fact:

"Alternatively, the discrepancies between the records could be due to production in the ice sheet resulting in significant amounts of excess COS in glacial period ice; this possibility was not considered by Aydin et al. (2016)."

Given that the correction is not intended to correct the spikes, the persistence of spikes after the correction cannot be perceived as evidence of production not happening. We cannot even rule out the possibility that the spikes represent atmospheric fluctuations. We state very clearly in the manuscript that the applied correction does not address high frequency variability because the same depth relationship is driven primarily by millennial scale variability (section 3.1). In multiple analyses scenarios presented in section 3.1, we also clearly demonstrate that the existence of spikes does not impact the applied correction. We could assume the spikes were also a result of production, eliminate them from the record by determining a baseline following some examples in literature, then conduct the exact same analyses presented in the manuscript and arrive at the same results, including the same-depth relationships driven by

millennial scale variability, and the same corrections for production that happens in the firn. The only difference would be that the corrected glacial period COS levels would be somewhat lower because the spikes would have been eliminated from the record before averaging. In essence, our approach of not trying to eliminate the spikes from the record is the more conservative approach.

The manuscript text about the 19 ky feature has been revised based on another comment by the reviewer (see below). As we note in the reply to that comment, we do not perceive a contradiction between the main point of the manuscript about lower atmospheric COS during the last glacial period and the implications of this observation on ocean productivity and whether the 19 ky feature is a real atmospheric event or not. Note that the 19 ky feature is delineated by 4 measurements out of 574.

Finally, the fact that the glacial period record may include some artifacts caused by production even after the correction does not invalidate the applied correction as an appropriate method. It only means that the glacial period COS levels could in fact be somewhat lower than what we present in the paper.

Delta age for the South Pole ice core during the glacial (1500 – 2700) is in the same range as peak-trough age separation of AIM (Antarctic isotope maximum) events. Is it possible, for example, that higher ssNa at AIM peaks correlates (imperfectly) in depth with higher COS at AIM troughs?
No, this is not possible. This should be evident from the SPC14 ssNa record plotted versus the composite Antarctic $CO_2$ record in Fig 4c. In a more general sense, any property measured in ice that correlates with ssNa will correlate with same-depth COS. The possible environmental causes of the ssNa variations, whether or not AIM evenst are relevant in this context, do not have any bearing on the interpretation of the the same-depth relationship between ssNa and COS. The same-age anticorrelation between ssNa and COS that emerges after the correction can be interpreted as a climate driven relationship. We focus solely on the glacial/interglacial change in the interpretation to keep the interpretation section focused on the most prominent feature of the record.

Alternatively, is it possible that there are multiple COS-altering mechanisms in the ice core, and in situ production related to organic S impurities that are correlated with ssNa deposition is happening at the same time as COS destruction by another process?
Yes, there are multiple mechanisms that alter COS in the ice cores. In addition to the production process, which is the main theme of this manuscript, COS undergoes in situ hydrolysis loss in ice cores. This fact is clearly acknowledged numerous times with relevant citations, including as early as L49-50 in the introduction: "Previous measurements of COS in Antarctic ice cores revealed slow, temperature-dependent degradation of COS in the ice core air due to hydrolysis (Aydin et al., 2014)." At lower ice sheet temperatures, hydrolysis loss is very slow and can practically be ignored at the South Pole. However, the WAIS Divide and Taylor Dome ice cores require a hydrolysis loss correction as well as accounting for the production (L267-270): "There are two other ice core COS records that extend back to the last glacial period. They are from the Taylor Dome (TD) and the West Antarctic Ice Sheet Divide (WD), Antarctica (Fig. 1c). Both of

these sites are warmer than the South Pole, therefore the TD and WD measurements require a correction for temperature-dependent hydrolysis loss (Aydin et al., 2014; 2016)."

If the reviewer is referring to some other process, they are not offering any specific evidence supporting this idea. It is impossible for us agree with or refute evidence that we do not see.

With regard to the inferred 2 – 4 times lower ocean COS source in the LGM as compared to the Holocene, I again think that this is a possible interpretation, but I am not convinced that this is the only possible interpretation. As the authors mention in their response, the evidence for ocean biological productivity changes during the deglaciation is mixed – there is paleoceanographic evidence for some regions being more productive during the LGM, and other regions less productive. Because of this, a scenario with no large ocean emission changes over the deglaciation (a scenario that would result if the in situ ssNa-based correction is not applied) seems possible to me.

We stated in the previous round of review that the glacial/interglacial change in global ocean productivity is an open science question, countering the reviewer's statement that there was plenty of evidence productivity was not lower during the last glacial period. We present the atmospheric COS record as an important piece of evidence that supports globally lower ocean productivity. Regional changes in ocean productivity are not relevant in the context atmospheric COS variability. As we stated in the previous round of the review, we offer an in depth discussion on this supported by plenty of citations (L527-596). Much like the first round of reviews, the reviewer does not directly challenge any specific evidence and arguments offered in the discussion section. If the reviewer choses to believe ocean productivity does not change between glacial and interglacial climates, that is their prerogative. We do not intend to convince everyone with one paper.

Related to this point is the large inferred COS peak at the end of LGM (≈19ka; COS on par with Holocene values), which is unexplained.

In the previous versions, we refrained from speculating about the nature of the positive excursion around 19 ky. We do not feel an obligation to offer an explanation for every feature in the record. Our inability to offer an explanation for any particular aspect of the record does not invalidate the explanations we offer for the data set as a whole.

That said, based on the current and the previous reviews, it appears that the reviewer is suggesting an interpretation of the 19 ky peak apparent in the SPC14 record as an atmospheric signal that may have resulted from an increase in ocean productivity. This is indeed possible, although we do not feel confident enough to make a strong claim about this without confirmation with measurements from other ice cores that there is indeed a peak at that time horizon. We would like to note that the current interpretation of lower biological productivity during the last glacial period does not in any way preclude the possibility of a relatively short-lived spike in ocean productivity during the LGM superimposed on the low baseline. We revised the relevant sections in the paper to clarify this and incorporate the reviewer's suggestion into the manuscript (L515-524):

"The 19 ky peak is characterized by four wet COS measurements and coincides in time with a shorter-lived sharp peak in ssNa (Fig. 4c). Given the prominence of spikes in the glacial period COS, we suspect that at least one of the measurements characterizing the 19 ky COS peak, possibly the highest measurement dating older than 20 ky, may be a coincidental, non-atmospheric spike while the other three measurements may characterize an atmospheric excursion of 50-100 ppt that is closer in duration to the peak in ssNa. This may explain why we do not see this feature in the WD COS record. An atmospheric COS excursion of this magnitude would represent a sudden and significant departure from the biogeochemical balance that maintains the low atmospheric COS levels during the LGM. Based solely on the magnitude, it could only be caused by an increase in ocean sulfur gas emissions or a decline in land biosphere uptake since these are the two major natural components of the COS budget (Table 1). This feature warrants further investigation if replicated with high resolution measurements from different ice cores."

Considering all of the above, my recommendation is still that the authors consider and include an alternative scenario in which the in situ correction is not applied.
As we stated in the beginning of our responses, we will not include an atmospheric interpretation of the uncorrected record, simply because we do not believe the measured COS values in the SPC14 ice core, particularly from the glacial period, represent atmospheric mixing ratios. We do not see any convincing evidence in the reviewer comments that suggests the opposite is true.

A more minor issue -- I am still not convinced that the in situ production the authors propose takes place mainly in the firn layer, for the reasons I highlighted in my original review. Wouldn't it also be possible, for example, that the concentration of impurities with depth that the authors mention would make production in deeper ice more likely than in shallow firn?
We are not sure what the reviewer means by concentration of impurities with depth. The impurity levels are higher in deeper ice from the last glacial period. This is not because impurities migrate in the ice sheet to deeper horizons after deposition. It is because surface snow and shallow firn were characterized by high impurity levels during the last glacial period.

But I agree with the authors that this process seems to be substrate-limited, otherwise there would a steady increase trend with depth as the authors suggest.

I think the authors have addressed my other comments sufficiently well.